# Adaptive Stochastic Coefficients for Accelerating Diffusion Sampling

**Ruoyu Wang**[1*]   **Beier Zhu**[1,2*]   **Junzhi Li**[3,4]   **Liangyu Yuan**[1,5]   **Chi Zhang**[1†]

[1]AGI Lab, Westlake University    [2]Nanyang Technological University
[3]University of Chinese Academy of Sciences
[4]Institute of Software, Chinese Academy of Sciences
[5] Tongji University
{wangruoyu71,chizhang}@westlake.edu.cn
beier.zhu@ntu.edu.sg
lijunzhi25@mails.ucas.ac.cn
liangyuy001@gmail.com

## Abstract

Diffusion-based generative processes, formulated as differential equation solving, frequently balance computational speed with sample quality. Our theoretical investigation of ODE- and SDE-based solvers reveals complementary weaknesses: ODE solvers accumulate irreducible gradient error along deterministic trajectories, while SDE methods suffer from amplified discretization errors when the step budget is limited. Building upon this insight, we introduce `AdaSDE`, a novel single-step SDE solver that aims to unify the efficiency of ODEs with the error resilience of SDEs. Specifically, we introduce a single per-step learnable coefficient, estimated via lightweight distillation, which dynamically regulates the error correction strength to accelerate diffusion sampling. Notably, our framework can be integrated with existing solvers to enhance their capabilities. Extensive experiments demonstrate state-of-the-art performance: at 5 NFE, `AdaSDE` achieves FID scores of $4.18$ on CIFAR-10, $8.05$ on FFHQ and $6.96$ on LSUN Bedroom. Codes are available in `https://github.com/Westlake-AGI-Lab/AdaSDE`.

## 1 Introduction

Diffusion Models (DMs) [1, 2, 3, 4, 5] have revolutionized generative modeling, achieving state-of-the-art performance across a broad range of applications [6, 7, 8, 9, 10, 11, 12, 13, 14]. Rooted in non-equilibrium thermodynamics, DMs learn to reverse a diffusion process: data are first gradually corrupted by Gaussian noise in a forward phase, and then reconstructed from pure noise through a learned reverse dynamics. This principled design offers stable training and exact likelihood modeling [15], resolving long-standing challenges in earlier approaches, *e.g.*, GANs [16] and VAEs [17].

Recent advances in diffusion models have highlighted the role of differential-equation solvers in balancing sampling speed and generation quality. We first develop a unified error analysis that decomposes the total approximation error into two orthogonal components: **(1) gradient error**, the discrepancy between the learned score function and the ground-truth score; and **(2) discretization error**, introduced by time discretization during sampling. Viewed through this lens, existing solvers exhibit complementary behaviors. *Ordinary differential equation (ODE)* based methods offer deterministic trajectories with modest discretization error for low number of function evaluations (NFEs), but their performance is fundamentally constrained by the irreversible accumulation of

---

[*]Equal contribution. [†]Corresponding author.

39th Conference on Neural Information Processing Systems (NeurIPS 2025).

gradient error [18, 19, 20, 21]. In contrast, *stochastic differential equation (SDE)* based methods inject stochasticity that can mitigate gradient error and enhance sample diversity; however, effectively suppressing gradient error in practice usually requires large step counts (*e.g.*, 100–1,000 NFEs) [2, 22]. Hybrid strategies such as restart sampling[23] alternate forward noise injection with backward ODE integration to combine these advantages, yet they still operate in high-NFE regimes.

Building on the above analysis, we introduce `AdaSDE`, a novel single-step SDE solver that unifies the computational efficiency of ODEs with the error resilience of SDEs under low-NFE budgets. Unlike traditional SDE solvers [24, 2] that inject fixed noise based on a predetermined time schedule, `AdaSDE` employs an *adaptive noise injection* mechanism controlled by a learnable stochastic coefficient $\gamma_i$ at each denoising step $i$. To effectively optimize $\gamma_i$, we further develop a *process-supervision* optimization framework that provides fine-grained guidance at each intermediate step rather than only supervising the final reconstruction. This design is inspired by the observation that diffusion trajectories exhibit consistent low-dimensional geometric structures across solvers and datasets [25, 26]. By aligning the geometry of the trajectories using $\gamma_i$, `AdaSDE` reduces gradient error through adaptive stochastic injection, while preserving deterministic efficiency of ODE solvers.

Extensive experiments on both pixel-space and latent-space DMs demonstrate the superiority of `AdaSDE`. Remarkably, with only 5 NFE, `AdaSDE` achieves FID scores of $4.18$ on CIFAR-10 [27] and $8.05$ on FFHQ $64\times64$ [28], surpassing the leading AMED-Solver [20] by $1.8\times$. Our contributions are threefold:

- We conduct a theoretical comparison of SDE and ODE error dynamics, demonstrating that SDEs offer more robust gradient error control.

- We introduce `AdaSDE`, the first single-step SDE solver that achieves efficient sampling ($<10$ NFEs) by optimizing adaptive $\gamma$-coefficients. Moreover, `AdaSDE` serves as a universal plug-in module that can enhance existing single-step solvers.

- Extensive evaluations on multiple generative benchmarks show that our `AdaSDE` achieves state-of-the-art performance with significant FID gains over existing solvers.

## 2   Related Work

Recent advancements in accelerating DMs primarily progress along two directions: improved numerical solvers and training-based distillation.

**Improved numerical solvers.** Early studies [2, 24] accelerated sampling by improving noise-schedule design, and DDIM [29] later introduced a non-Markovian formulation that enabled deterministic and much faster sampling. The establishment of the probability-flow ODE view [15] further unified diffusion formulations and paved the way for higher-order numerical schemes and practical preconditioning strategies, exemplified by EDM [30]. Following this insight, a series of ODE/SDE integrators have emerged to push the accuracy–speed frontier. For instance, DEIS [31], DPM-Solver [21], and DPM-Solver++[22] exploit exponential integration, Taylor expansion, and data-prediction parameterization to achieve robust few-step sampling. Linear multistep variants, including iPNDM [32, 31] and UniPC [33], further enable efficient DMs sampling with $\sim10$ NFE. Hybrid and stochastic extensions extend beyond deterministic solvers: Restart Sampling [23] alternates ODE trajectories with SDE-style noise injection; SA-Solver [34] introduces a training-free stochastic Adams multi-step scheme with variance-controlled noise.

**Training-based distillation.** Two main paradigms dominate this research direction. The first is *trajectory approximation*, which uses compact student networks to approximate trajectories generated by teacher models, reducing computational steps. This can be achieved offline: by curating datasets from pre-generated samples [35], or online through progressive distillation that gradually decreases the number of sampling steps [36, 18]. The second paradigm is *temporal alignment*, which enforces coherence across sampling trajectories by aligning intermediate predictions between adjacent timesteps [37, 38], or by minimizing distributional gaps between real and synthesized data [39, 40]. While these methods improve generation quality and efficiency, they typically require substantial computational resources and complex training protocols, limiting their practicality. Recent distillation-based solvers—such as AMED [20], EPD [41], and D-ODE [37]—achieve few-step sampling through lightweight tuning rather than full retraining. Complementary efforts on time schedule optimization, including LD3 [42], DMN [43], and GITS [26], further improve efficiency. While most few-step

samplers are rooted in ODE formulations, our approach introduces few-step SDE-driven generation by learning stochastic coefficients under a computationally lightweight objective.

# 3 Preliminaries

## 3.1 Diffusion Models with Differential Equations

DMs define a forward process that perturbs data into a noise distribution, followed by a learned reverse process that inverts this perturbation to generate samples. The forward process is designed as a stochastic trajectory governed by a predefined noise schedule, which can be described by:

$$\mathrm{d}\mathbf{x} = \frac{\dot{s}(t)}{s(t)}\mathbf{x} + s(t)\sqrt{2\sigma(t)\dot{\sigma}(t)}\mathrm{d}\mathbf{w} \tag{1}$$

where $\sigma(t)$ is the monotonically increasing noise schedule, and $\mathbf{w}$ denotes a standard Wiener process. This formulation ensures that the marginal distribution $p_t(\mathbf{x})$ at time $t$ corresponds to the convolution of the data distribution $p_0 = p_{\text{data}}$ with a Gaussian kernel of variance $\sigma^2(t)$. By selecting a sufficiently large terminal time $T$, $p_T$ converges to an isotropic Gaussian $\mathcal{N}(\mathbf{0}, \sigma^2(T)\mathbf{I})$, serving as the prior. Sampling is performed by reversing the forward dynamics through either a reverse-time SDE in Eq. (1) or an ODE [15]:

$$\mathrm{d}\mathbf{x} = -\sigma(t)\dot{\sigma}(t)\nabla_{\mathbf{x}}\log p_t(\mathbf{x})\mathrm{d}t. \tag{2}$$

Here, the score function $\nabla_{\mathbf{x}}\log p_t(\mathbf{x})$ is the drift term that guides samples toward high density regions of $p_0$. Following common practice [19], the noise schedule is simplified to $\sigma(t) = t$, reducing $\sigma(t)\dot{\sigma}(t)$ to $t$. A neural network $s_\theta(\mathbf{x}, t)$ is optimized through denoising score matching [15] to estimate the score function. The training objective minimizes the weighted expectation:

$$\mathbb{E}_{t,\mathbf{x}_0,\mathbf{x}_t}\left[\lambda(t)\|s_\theta(\mathbf{x}_t, t) - \nabla_{\mathbf{x}_t}\log p_t(\mathbf{x}_t \mid \mathbf{x}_0)\|^2\right] \tag{3}$$

where $\lambda(t)$ specifies the loss weighting schedule and $p_t(\mathbf{x}_t \mid \mathbf{x}_0)$ denotes the Gaussian transition kernel of the forward process. During sampling, $s_\theta(\mathbf{x}, t)$ serves as a surrogate for the true score in the reverse-time dynamics, reducing the general SDE in Eq. (2) to the deterministic gradient flow:

$$\mathrm{d}\mathbf{x} = s_\theta(\mathbf{x}_t, t)\mathrm{d}t \tag{4}$$

# 4 Analysis of ODE and SDE

## 4.1 Trade-offs Between ODE and SDE Solvers

The choice between ODE and SDE solvers in DMs entails trade-offs among sampling speed, quality, and error dynamics. ODE solvers, characterized by deterministic trajectories, offer computational efficiency and stability through compatibility with compatibility with higher-order numerical methods, *e.g.*, iPNDM [32, 31]. Such solvers reduce local discretization errors and achieve competitive sample quality with as few as 10–50 steps [21, 19]. However, their deterministic nature limits their ability to correct errors from imperfect score function approximations, leading to performance plateaus as step count increases [23]. Furthermore, the absence of stochasticity may suppress fine-grained variations, potentially reducing sample diversity compared to SDE-based methods [2].

In contrast, SDE solvers leverage stochasticity to counteract accumulated discretization and gradient errors over time, enabling superior sample fidelity in high-step regimes [23]. The injected noise further encourages exploration of the data manifold, improving diversity [2]. However, these benefits come at the cost of significantly larger step counts (typically 100–1,000) required to suppress errors that scale as $O(\delta^{3/2})$, compared to $O(\delta^2)$ for ODEs [23, 44]. Moreover, SDE trajectories are highly sensitive to suboptimal noise schedules, particularly in low-step settings [24]. While reverse-time SDEs theoretically guarantee convergence to the true data distribution under ideal conditions [45], their computational cost often renders them impractical for real-time applications.

Recent hybrid approaches, such as Restart sampling [23], reconcile these trade-offs by alternating deterministic steps with stochastic resampling, leveraging ODE efficiency for coarse trajectory simulation while resetting errors via SDE-like noise injection. This strategy highlights the complementary strengths of both methods, positioning hybrid frameworks at the forefront of quality-speed Pareto frontiers in diffusion-based generation. However, Restart sampling still performs under high-step regimes (>50 steps).

## 4.2 Error Propagation in Deterministic and Stochastic Sampling

The trade-offs discussed in Section 4.1 raise a key question:

*Can SDE-based approaches achieve efficient sampling with substantially fewer steps?*

To answer this, we build on the theoretical frameworks of [23, 44] to analyze the total sampling error of ODE and SDE formulations under the Wasserstein-1 metric. We begin with the discretized ODE system $\text{ODE}_\theta$, governed by the learned drift field $s_\theta$, and examine its approximation behavior over the interval $[t, t + \Delta t] \subset [0, T]$. Theorem 1 formalizes this analysis and establishes an upper bound on the Wasserstein-1 distance between the generated and true data distributions (proof in Appendix B.1).

**Theorem 1.** *(ODE Error Bound [23]) Let $\Delta t > 0$ denote the discretization step size. Over the interval $[t, t + \Delta t]$, the trajectory $\mathbf{x}_t = \text{ODE}_\theta (\mathbf{x}_{t+\Delta t}, t + \Delta t \to t)$ is generated by the learned drift $s_\theta$, and the induced distribution is denoted by $p_t^{\text{ODE}_\theta}$. We make the following assumptions:*
**A1. Lipschitz and bounded drift:** *$ts_\theta(\mathbf{x}, t)$ is $L_2$-Lipschitz in $\mathbf{x}$, $L_0$-Lipschitz in $t$ and uniformly bounded by $L_1$.*
**A2:** *The learned drift satisfies a uniform supremum bound:* $\sup_{\mathbf{x}, t} \|ts_\theta(\mathbf{x}, t) - t\nabla \log p_t(\mathbf{x})\| \leq \epsilon_t$.
**A3. Bounded trajectories:** *$\|\mathbf{x}_t\| \leq B/2$ for all $t \in [t, t + \Delta t]$.*
*The Wasserstein-1 distance between $p_t^{\text{ODE}_\theta}$ and the true distribution $p_t$ satisfies:*

$$\underbrace{W_1\left(p_t^{\text{ODE}_\theta}, p_t\right)}_{\text{total error}} \leq \underbrace{B \cdot \text{TV}\left(p_{t+\Delta t}^{\text{ODE}_\theta}, p_{t+\Delta t}\right)}_{\text{① gradient error bound}} + \underbrace{e^{L_2 \Delta t}\left(\Delta t(L_2 L_1 + L_0) + \epsilon_t\right)\Delta t}_{\text{② discretization error bound}} \tag{5}$$

*where $\text{TV}(\cdot, \cdot)$ denotes the total variation distance.*

The bound in Eq. 5 consists of two term distinct interpretations. The first term ① is the gradient error bound which reflects the discrepancy between the learned score function and the ground-truth one at the start time $t + \Delta t$. It also captures the propagation of errors accumulated from earlier time steps. The second term ② is the discretization error bound, which represents the newly introduced errors within the current interval $[t, t + \Delta t]$. Since the ODE process is deterministic, any discrepancy between the generated and true distributions at $t + \Delta t$ is directly carried forward to time $t$, without stochastic mechanisms to dissipate it.

Next, we introduce our $\texttt{AdaSDE}$ update over the interval $[t, t + \Delta t]$, defined as:

$$\mathbf{x}_t = \texttt{AdaSDE}_\theta(\mathbf{x}_{t+\Delta t}, t + \Delta t \to t, \gamma),$$

which inserts a stochastic *forward perturbation* followed by a deterministic *backward process*.

$$\mathbf{x}_{t+\Delta t}^\gamma = \mathbf{x}_{t+(1+\gamma)\Delta t} = \mathbf{x}_{t+\Delta t} + \varepsilon_{t+\Delta t \to t+(1+\gamma)\Delta t}, \qquad \text{(Forward process)}$$
$$\mathbf{x}_t = \text{ODE}_\theta\left(\mathbf{x}_{t+\Delta t}^\gamma, t + (1+\gamma)\Delta t \to t\right), \qquad \text{(Backward process)}$$

where

$$\varepsilon_{t+\Delta t \to t+(1+\gamma)\Delta t} \sim \mathcal{N}\left(\mathbf{0}, \left((t + (1+\gamma)\Delta t)^2 - (t + \Delta t)^2\right)\mathbf{I}\right).$$

Here, $\gamma \in (0, 1)$ is a tunable coefficient with its optimization deferred in Section 5. Different from deterministic ODE, $\texttt{AdaSDE}$ introduces controlled noise injection to mitigate error accumulation. Theorem 2 establishes an error bound between the generated and the true data distribution for our $\texttt{AdaSDE}$ (proof in Appendix B.2).

**Theorem 2.** *Under the same assumptions in Theorem 1. Let $p_t^{\text{AdaSDE}_\theta}$ denote the distribution after* $\texttt{AdaSDE}$ *update over the interval $[t, t + \Delta t]$. Then*

$$W_1\left(p_t^{\text{AdaSDE}_\theta}, p_t\right) \leq \underbrace{B \cdot (1 - \lambda(\gamma))\text{TV}\left(p_{t+(1+\gamma)\Delta t}^{\text{AdaSDE}}, p_{t+(1+\gamma)\Delta t}\right)}_{\text{gradient error bound}} \tag{6}$$

$$+ \underbrace{e^{(1+\gamma)L_2 \Delta t}(1 + \gamma)\left((1+\gamma)\Delta t\left(L_2 L_1 + L_0\right) + \epsilon_t\right)\Delta t}_{\text{discretization error bound}} \tag{7}$$

*where $\lambda(\gamma) = 2Q\left(\dfrac{B}{2\sqrt{(t + (1+\gamma)\Delta t)^2 - t^2}}\right)$, $Q(r) = \Pr(a \geq r)$ for $a \sim \mathcal{N}(0, 1)$.*

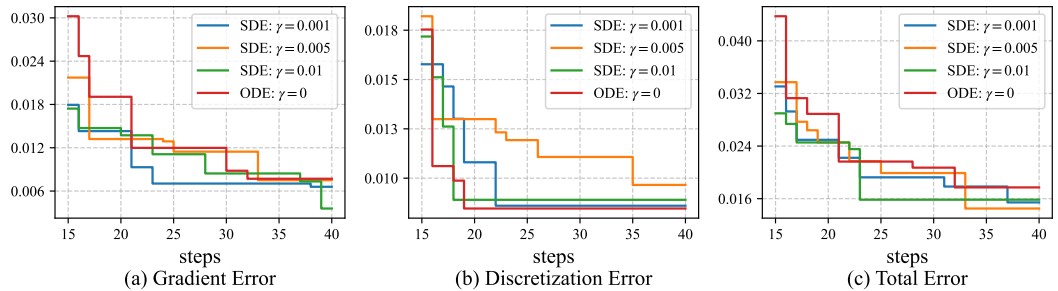

| (a) Gradient Error | (b) Discretization Error | (c) Total Error |

Figure 1: Gradient error, Discretization error and Total error on synthetic dataset across various steps (measured in 1-Wasserstein Distance). $\gamma = 0$ indicates adding no stochasticity (ODE), $\gamma > 0$ indicates SDE variants, figures are plotted in Pareto Frontier.

As shown in Theorem 2, the decoupled formulation tightens the Wasserstein-1 error bound through a reduced coefficient $B(1-\lambda(\gamma))$. We next formalize this improvement by comparing the gradient-error terms of ODE and `AdaSDE` formulations in Theorem 3.

**Theorem 3.** *Under the same assumptions as in Theorem 1 and Theorem 2, we denote:*

$$\mathcal{E}_{\text{grad}}^{\text{ODE}} = B \cdot \text{TV}\Big(p_{t+\Delta t}^{\text{ODE}_\theta}, \, p_{t+\Delta t}\Big), \qquad \text{(ODE gradient error)}$$

$$\mathcal{E}_{\text{grad}}^{\text{AdaSDE}} = B \cdot \big(1 - \lambda(\gamma)\big) \, \text{TV}\Big(p_{t+(1+\gamma)\Delta t}^{\text{AdaSDE}}, \, p_{t+(1+\gamma)\Delta t}\Big). \qquad \text{(SDE gradient error)}$$

*Then we have $\mathcal{E}_{\text{grad}}^{\text{AdaSDE}} \leq \mathcal{E}_{\text{grad}}^{\text{ODE}}$, where the inequality is strict when $\gamma > 0$.*

*Proof sketch.* (full proof in Appendix B.3) For the ODE update, $\mathcal{E}_{\text{grad}}^{\text{ODE}}$ depends on the total-variation distance between the distributions at time $t + \Delta t$. For `AdaSDE` update, $\mathcal{E}_{\text{AdaSDE}}$ includes a contraction factor $(1 - \lambda(\gamma))$ and is evaluated at the higher noise level $t + (1+\gamma)\Delta t$. Define the Gaussian kernel

$$\phi_\gamma(z) = (2\pi\sigma_\gamma^2)^{-d/2} \exp\left(-\frac{\|z\|^2}{2\sigma_\gamma^2}\right), \qquad \sigma_\gamma^2 = \big(t + (1+\gamma)\Delta t\big)^2 - \big(t + \Delta t\big)^2.$$

The distributions after the noise injection satisfy

$$p_{t+(1+\gamma)\Delta t} = p_{t+\Delta t} * \phi_\gamma, \qquad q_{t+(1+\gamma)\Delta t} = q_{t+\Delta t} * \phi_\gamma.$$

By Lemma 6 in Appendix, convolution with the same Gaussian kernel does not increase total variation distance:

$$\text{TV}(p_{t+\Delta t} * \phi_\gamma, \, q_{t+\Delta t} * \phi_\gamma) \leq \text{TV}(p_{t+\Delta t}, \, q_{t+\Delta t}).$$

Consequently,

$$\mathcal{E}_{\text{grad}}^{\text{AdaSDE}} \leq (1 - \lambda(\gamma)) \, \mathcal{E}_{\text{grad}}^{\text{ODE}},$$

with a strictly smaller bound whenever $\gamma > 0$. $\qquad\square$

Although the gradient error term of `AdaSDE` enjoys a tighter bound through $B(1 - \lambda(\gamma))$, its discretization error grows rapidly under large time steps $(\Delta t)$ with noise schedules scaling as $\gamma(t) \propto \Delta t$. Specifically, the exponential growth factor $e^{(1+\gamma)L_2\Delta t}$ combined with the quadratic $\Delta t$-dependence in $(1 + \gamma)^2 \Delta t^2 (L_2 L_1 + L_0)$ creates error amplification that scales asymptotically as $O(\Delta t e^{C\Delta t})$ when $\gamma \sim O(\Delta t)$. This dominates the improved gradient error control, particularly during critical initial denoising steps where the product $(1 + \gamma)\Delta t$ violates discretization stability conditions. This amplification offsets the benefit of gradient-error contraction, causing total error accumulation along the trajectory and explaining the degraded few-step performance of SDE-based sampling in practice.

### 4.3 Synthetic Validation

To verify the error-mitigation capability of stochastic updates in `AdaSDE`, we conduct experiments on a 2D double-circle synthetic dataset, comparing the total, gradient, and discretization errors.

**Setup.** As illustrated in Figure 2, we use a 2D double-circle dataset consisting of $20,000$ samples uniformly distributed along two concentric circles with radii of $0.8$ (outer) and $0.6$ (inner), each perturbed by Gaussian noise with a standard deviation of $0.1$. We follow the training and sampling procedures of EDM [30] to model the data distribution, employing the second-order Heun method for SDE integration. The stochastic coefficient $\gamma$ is varied over $\{0, 0.001, 0.005, 0.01\}$, where $\gamma = 0$ corresponds to the deterministic ODE sampler.

To quantify different types of errors, we measure the 2D Wasserstein-1 distance between corresponding distributions. **The total error** is computed as the distance between the ground-truth data distribution and the generated distribution. To estimate **gradient** and **discretization errors**, we first construct an intermediate regenerated distribution. Specifically, given the dataset of $20,000$ samples, we perturb each point by Gaussian noise according to

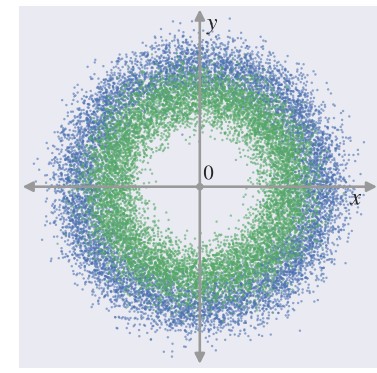

Figure 2: Illustration of the 2D double-circles: two concentric rings with radii $0.8$ (outer, blue) and $0.6$ (inner, green). We uniformly sample 20,000 points and add isotropic Gaussian noise ($\sigma = 0.1$).

$\mathbf{x}_{t_{\text{mid}}} = \mathbf{x}_0 + t_{\text{mid}}\sigma$, where $t_{\text{mid}} = 0.8$ and perform one-third of a denoising step to obtain the regenerated samples. The gradient error is defined as the distance between the regenerated distribution and the model-generated distribution at $T = 80.0$, while the discretization error is defined as the distance between the regenerated distribution and the ground-truth distribution.

**Result.** The gradient error, discretization error, and total error over the steps range $t \in [15, 40]$ are illustrated in Figure 1. It is observed that the discretization error of ODEs is less than that of SDE variants (in Figure 1 (b)), corresponding to the derived result that the upper bound for ODE sampling error (stated in Theorem 1) is less than that for SDEs (stated in Theorem 2) by a multiplication factor. However, the gradient error (*i.e.*, error caused by network approximation) of SDEs ($\gamma > 0$) drops compared to ODE counterparts (in Figure 1 (a)), validating the Wasserstein-1 distance bound in Theorem 3. The stochastic step is effective in alleviating the gradient error made by network approximation. Consequently, as shown in Figure 1 (c), the total error accumulated throughout the sampling process decreases due to the reduction of gradient error brought by stochasticity, confirming the effectiveness of our approach in improving sampling accuracy. Given the above theoretical analysis and synthetic validation on Wasserstein-1 distance, we present the following remark.

**Remark 1.** *Let $\mathcal{E}_{\text{total}}(N, \gamma)$ represent the accumulated sampling error for a discretization of $N$ steps with parameter $\gamma$. Then for $\forall N \in \mathbb{Z}^+$, $\exists \gamma \in (0, 1)$ such that:*

$$\mathcal{E}_{\text{total}}(N, \gamma) \leq \mathcal{E}_{\text{total}}(N, 0)$$

## 5 Methodology

Building on the above theoretical and empirical validation, we introduce `AdaSDE`, a single-step SDE solver that parameterizes the stochastic coefficient $\gamma$ as learnable variable. This design unleashes the potential of SDE-based solvers under low-NFE regimes.

### 5.1 Sampling Trajectory Geometry

The trajectories generated by Eq. (4) exhibit low complexity geometric features with implicit connections to annealed mean displacement, as established in previous work [26, 25]. Each sample initialized from the noise distribution progressively approaches the data manifold through smooth, quasi-linear trajectories characterized by monotonic likelihood improvement. In addition, under identical dataset and time schedule, all sampling trajectories demonstrate geometric consistency across different sampling methods. This geometric insight motivates a discrete-time distillation framework. By strategically inserting intermediate temporal steps within student trajectories, we construct high-fidelity reference trajectories. This enables process-supervised optimization that rigorously determines the governing $\gamma$ parameters for trajectory segments. Specifically, given a student time schedule $\mathcal{T}_{\text{stu}} = \{t_0, t_1, \ldots, t_N\}$ with $N$ steps, we insert $M$ intermediate steps between $t_n$ and $t_{n+1}$ (denoted as $\mathcal{T}_{\text{tea}} = \{t_0, t_0^{(1)}, \ldots, t_0^{(M)}, t_1, \ldots, t_N\}$ ) to generate refined teacher trajectories. Notably,

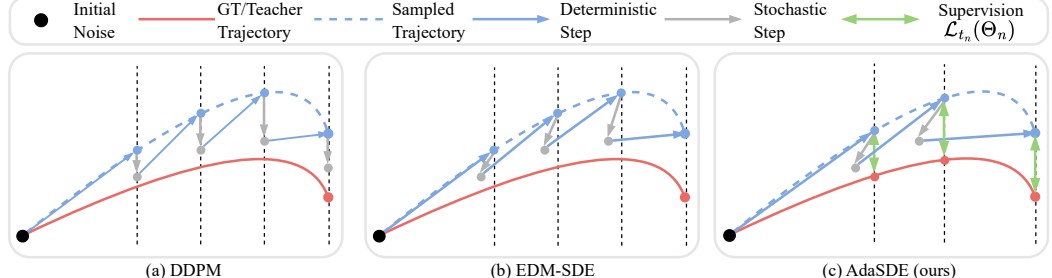

Figure 3: The proposed `AdaSDE` framework. `AdaSDE` diverges from traditional heuristic noise injection methods used in DDPM [2] and EDM-SDE [19]. Instead, we use intermediary supervision from a teacher sampling path to learn and optimize the noise injection process.

---

**Algorithm 1** Optimizing $\Theta_{1:N}$

1: **Given:** time schedules $\mathcal{T}_{\text{stu}}$ and $\mathcal{T}_{\text{tea}}$
2: **Repeat until** convergence
3:     Sample $\mathbf{x}_{t_N} = \mathbf{y}_{t_N} \sim \mathcal{N}(\mathbf{0}, t_N^2\mathbf{I})$
4:     **for** $n = N$ **to** 1 **do**
5:         Sample $\boldsymbol{\epsilon}_n \sim \mathcal{N}(\mathbf{0}, \mathbf{I})$
6:         $\{\gamma, \xi, \lambda, \mu\}_n \leftarrow \Theta_n$
7:         $\hat{t}_n \leftarrow t_n + \gamma_n t_n$
8:         $\mathbf{x}_{t_n} \leftarrow \mathbf{x}_{t_n} + \sqrt{\hat{t}_n^2 - t_n^2}\,\boldsymbol{\epsilon}_n$
9:         Compute $\mathbf{x}_{t_{n-1}}$ using Eq. (9)
10:        Update $\Theta_n$ via Eq. (10)
11:    **end for**

---

**Algorithm 2** `AdaSDE` sampling

1: **Given:** parameters $\Theta_{1:N}$, student time schedule $\mathcal{T}_{\text{stu}}$
2: Initialize $\mathbf{x}_{t_N} \sim \mathcal{N}(\mathbf{0}, t_N^2\mathbf{I})$
3: **for** $n = N$ **to** 1 **do**
4:     Sample $\boldsymbol{\epsilon}_n \sim \mathcal{N}(\mathbf{0}, \mathbf{I})$
5:     $\{\gamma, \xi, \lambda, \mu\}_n \leftarrow \Theta_n$
6:     $\hat{t}_n \leftarrow t_n + \gamma_n t_n$
7:     $\mathbf{x}_{t_n} \leftarrow \mathbf{x}_{t_n} + \sqrt{\hat{t}_n^2 - t_n^2}\,\boldsymbol{\epsilon}_n$
8:     Compute $\mathbf{x}_{t_{n-1}}$ using Eq. (9)
9: **end for**
10: **Return:** $\mathbf{x}_{t_0}$

---

our interpolation scheme employs a flexible strategy that allows for selecting different time schedules based on various solvers. This adaptability enhances the fidelity of teacher trajectories.

## 5.2 Fast SDE-based Sampling

We extend the midpoint-based correction mechanisms Eq. (8) from AMED-Solver [20] to SDEs, proposing a sampling framework that strategically aligns stochastic perturbations with learned trajectory geometry.

$$\mathbf{x}_{t_n} \approx \mathbf{x}_{t_{n+1}} + (t_n - t_{n+1}) \underbrace{s_\theta\left(\mathbf{x}_{\xi_n}, \xi_n\right)}_{\text{midpoint gradient}}, \xi_n \in [t_{n+1}, t_n] \tag{8}$$

The parameterization adopts the design from DPM-Solver's intermediate time step construction, formally defined as $\xi_n = \sqrt{t_n t_{n+1}}$. This square-root formulation guarantees smooth geometric interpolation between adjacent time steps in the noise scheduling process. Building on insights from [46, 47] showing network scaling mitigates input mismatches, we propose learnable parameters $\{\lambda_n, \mu_n\}$ to adaptively adjust both exposure bias and timestep scales. The parameters $\Theta_n = \{\gamma_n, \xi_n, \lambda_n, \mu_n\}_{n=1}^N$ are optimized through our discrete-time distillation framework described in Section 5.1. Consequently, Eq. (8) can be reformulated in the following form:

$$\mathbf{x}_{t_n} \approx \mathbf{x}_{t_{n+1}} + (1 + \lambda_n)(t_n - t_{n+1}) s_\theta\left(\mathbf{x}_{\xi_n}, \xi_n + \mu_n\right) \tag{9}$$

Let $\{\mathbf{y}_{t_n}\}_{n=1}^N$ denote the reference states of teacher trajectories. Starting from the identical initial noise $\mathbf{y}_{t_0}$, we generate student trajectories by optimizing the parameter sequence $\{\Theta_n\}_{n=1}^N$, resulting in student states $\{\mathbf{x}_{t_n}\}_{n=1}^N$ that align with the teacher trajectories under a predefined metric $d(\cdot, \cdot)$. Crucially, since $\mathbf{x}_{t_n}$ depends on all preceding parameters $\{\Theta_n\}_{n=1}^N$ through the iterative sampling process, we implement stagewise optimization by minimizing the cumulative alignment loss at each timestep $t_n$:

$$\mathcal{L}_{t_n}(\Theta_n) = d\left(\mathbf{x}_{t_n}, \mathbf{y}_{t_n}\right) \tag{10}$$

Table 1: Image generation results across different datasets. (a) CIFAR10 [35] (unconditional), (b) FFHQ [28] (unconditional), (c) ImageNet [49] (conditional), (d) LSUN Bedroom [50] (unconditional). We compared AdaSDE-Solver and the training-required method AMED-Solver [20], as well as other training-free methods. AdaSDE achieves superior performance across all datasets.

(a) CIFAR10 $32 \times 32$ [27]

| Method | NFE | | | |
|---|---|---|---|---|
| | 3 | 5 | 7 | 9 |
| Multi-Step Solvers | | | | |
| DPM-Solver++(3M) [22] | 110.0 | 24.97 | 6.74 | 3.42 |
| UniPC [33] | 109.6 | 23.98 | 5.83 | 3.21 |
| iPNDM [32, 31] | 47.98 | 13.59 | 5.08 | 3.17 |
| Single-Step Solvers | | | | |
| DDIM [29] | 93.36 | 49.66 | 27.93 | 18.43 |
| Heun [19] | 306.2 | 97.67 | 37.28 | 15.76 |
| DPM-Solver-2 [21] | 153.6 | 43.27 | 16.69 | 8.65 |
| DPM-Plugin (ours) | 39.57 | 13.75 | 9.19 | 7.21 |
| AMED-Solver [20] | 18.49 | 7.59 | 4.36 | 3.67 |
| AdaSDE (ours) | **12.62** | **4.18** | **2.88** | **2.56** |

(c) ImageNet $64 \times 64$ [49]

| Method | NFE | | | |
|---|---|---|---|---|
| | 3 | 5 | 7 | 9 |
| Multi-Step Solvers | | | | |
| DPM-Solver++(3M) [22] | 91.52 | 25.49 | 10.14 | 6.48 |
| UniPC [33] | 91.38 | 24.36 | 9.57 | 6.34 |
| iPNDM [32, 31] | 58.53 | 18.99 | 9.17 | 5.91 |
| Single-Step Solvers | | | | |
| DDIM [29] | 82.96 | 43.81 | 27.46 | 19.27 |
| Heun [19] | 249.4 | 89.63 | 37.65 | 16.76 |
| DPM-Solver-2 [21] | 140.2 | 59.47 | 22.02 | 11.31 |
| DPM-Plugin (ours) | 108.9 | 17.03 | 11.69 | 8.06 |
| AMED-Solver [20] | 38.10 | 10.74 | 6.66 | 5.44 |
| AdaSDE (ours) | **18.51** | **6.90** | **5.26** | **4.59** |

(b) FFHQ $64 \times 64$ [28]

| Method | NFE | | | |
|---|---|---|---|---|
| | 3 | 5 | 7 | 9 |
| Multi-Step Solvers | | | | |
| DPM-Solver++(3M) [22] | 86.45 | 22.51 | 8.44 | 4.77 |
| UniPC [33] | 86.43 | 21.40 | 7.44 | 4.47 |
| iPNDM [32, 31] | 45.98 | 17.17 | 7.79 | 4.58 |
| Single-Step Solvers | | | | |
| DDIM [29] | 78.21 | 43.93 | 28.86 | 21.01 |
| Heun [19] | 356.5 | 116.7 | 54.51 | 28.86 |
| DPM-Solver-2 [21] | 215.7 | 74.68 | 36.09 | 16.89 |
| DPM-Plugin (ours) | 66.31 | 20.80 | 14.51 | 10.48 |
| AMED-Solver [20] | 47.31 | 14.80 | 8.82 | 6.31 |
| AdaSDE (ours) | **23.80** | **8.05** | **5.11** | **4.19** |

(d) LSUN Bedroom $256 \times 256$ [50]

| Method | NFE | | | |
|---|---|---|---|---|
| | 3 | 5 | 7 | 9 |
| Multi-Step Solvers | | | | |
| DPM-Solver++(3M) [22] | 111.9 | 23.15 | 8.87 | 6.45 |
| UniPC [33] | 112.3 | 23.34 | 8.73 | 6.61 |
| iPNDM [32, 31] | 80.99 | 26.65 | 13.80 | 8.38 |
| Single-Step Solvers | | | | |
| DDIM [29] | 86.13 | 34.34 | 19.50 | 13.26 |
| Heun [19] | 291.5 | 175.7 | 78.66 | 35.67 |
| DPM-Solver-2 [21] | 227.3 | 47.22 | 23.21 | 13.80 |
| DPM-Plugin (ours) | 97.13 | 21.02 | 13.68 | 10.89 |
| AMED-Solver [20] | 58.21 | 13.20 | 7.10 | 5.65 |
| AdaSDE (ours) | **18.03** | **6.96** | **5.69** | **5.16** |

In each training loop, we perform backpropagation $N$ times. The comparison with existing SDE solvers are presented in Figure 3. The complete training algorithm is detailed in Algorithm 1, while the inference procedure is outlined in Algorithm 2. AdaSDE serves as a plug-and-play module for existing solvers. To implement this, we train the AdaSDE predictor Algorithm 1 by replacing the mean update in Equation (8) with the target solver's formulation.

# 6 Experiments

## 6.1 Experiment Setup

**Models and datasets.** We apply `AdaSDE` and DPM-Plugin to five pre-trained diffusion models across diverse domains. For pixel-space models, we include CIFAR10 ($32 \times 32$) [27], FFHQ ($64 \times 64$) [48], and ImageNet ($64 \times 64$) [49]. For latent-space models, we evaluate LSUN Bedroom ($256 \times 256$) [50] with a guidance scale of 1.0. Additionally, we consider text-to-image high-resolution generation models, including Stable Diffusion v1.5 [5] at $512 \times 512$ resolution with a guidance scale of 7.5.

**Solvers and time schedules.** We compare `AdaSDE` against state-of-the-art single-step and multi-step ODE solvers. The single-step baselines include training-free methods—DDIM [29], EDM [19], and DPM-Solver-2 [21], as well as the lightweight-tuning approach AMED-Solver [20]. For multi-step methods, we evaluate DPM-Solver++ (3M) [22], UniPC [33], and iPNDM [32, 31]. To further demonstrate the effectiveness of our method, we also conduct a head-to-head comparison between DPM-Plugin and DPM-Solver-2 [21].

Table 2: FID results on Stable Diffusion v1.5 [5] with a classifier-free guidance weight $w = 7.5$.

| Method | NFE | | | |
|---|---|---|---|---|
| | 4 | 6 | 8 | 10 |
| **MSCOCO 512×512** | | | | |
| DPM-Solver++(2M) [22] | 21.33 | 15.99 | 14.84 | 14.58 |
| AMED-Plugin [20] | **18.92** | 14.84 | 13.96 | 13.24 |
| DPM-Solver-v3 [51] | - | 16.41 | 15.41 | 15.32 |
| AdaSDE (ours) | 30.89 | **13.99** | **13.39** | **12.68** |

Table 3: Ablation study of time schedules on CIFAR-10 [27].

| Time schedule | NFE | | | |
|---|---|---|---|---|
| | 3 | 5 | 7 | 9 |
| **CIFAR-10 32×32** | | | | |
| Time Uniform [2] | 12.62 | **4.18** | **2.88** | **2.56** |
| Time Polynomial [19] | **11.61** | 10.05 | 5.14 | 3.35 |
| Time LogSNR [21] | 23.38 | 10.42 | 7.96 | 4.84 |

To ensure an equitable and consistent comparison, our study faithfully adheres to the time scheduling strategies as recommended in the related work [19, 22, 33]. Specifically, we implement the logarithmic signal-to-noise ratio (logSNR) scheduling for DPM-Solver{-2, ++(3M)} and UniPC algorithms. For other baseline algorithms, EDM time schedule with $\rho$ set to 7 has been employed. For AdaSDE and DPM-Plugin, we implement time-uniform schedule.

**Learned perceptual image patch similarity** While some search-based frameworks employ LPIPS as their distance metric [52], we observed that using LPIPS during the intermediate steps of our method provided no significant performance gains and substantially increased training duration. Consequently, to balance efficiency and final quality, our approach utilizes Mean Squared Error (MSE) for optimizing intermediate steps, while applying the LPIPS metric in the final stage to enhance the overall training outcome.

**Training details.** Our AdaSDE is assessed at low NFE settings (NFE $\in \{3, 5, 7, 9\}$) with AFS [53] implemented. Sample quality is gauged using the Fréchet Inception Distance (FID) [54] over 50k images. For Stable-Diffusion, We evaluate FID as [54], using 30k samples from fixed prompts based on the MS-COCO [28] validation set. The random seed was fixed to 0 to ensure consistent reproducibility of the experimental results.

## 6.2 Main Results

In table 1, we benchmark AdaSDE against single- and multi-step baseline solvers on CIFAR-10, FFHQ, ImageNet 64×64, and LSUN Bedroom across varying NFE. We observe *consistent and substantial* improvements in the low-step regime (3–9 NFE). For example, at NFE=9 we obtain FIDs of 4.59 (ImageNet) and 5.16 (LSUN Bedroom), while the second-best single-step baseline (AMED-Solver) reaches 5.44 and 5.65, respectively, indicating clear gains. In an even more challenging few-step setting (NFE=3 on LSUN Bedroom), AdaSDE achieves 18.03 FID, markedly outperforming AMED-Solver's 58.21. On CIFAR-10, NFE=5 yields 4.18 FID (vs. AMED-Solver's 7.59); on FFHQ, NFE=5 yields 8.05, substantially better than DPM-Plugin's 20.80 and DPM-Solver-2's 74.68. Overall, AdaSDE maintains—and often widens—its advantage as the number of steps decreases.

We further evaluate AdaSDE on Stable Diffusion v1.5 with classifier-free guidance set to 7.5, reporting FID on the MS-COCO validation set (see table 2). At NFE=8/10, AdaSDE attains 13.39/12.68, surpassing DPM-Solver++(2M) at 14.84/14.58 and AMED-Plugin at 13.96/13.24, while remaining competitive with DPM-Solver-v3 across multiple step counts. These results indicate that our adaptive stochastic coefficient not only improves pixel-space diffusion models but also transfers robustly to high-resolution text-to-image generation in latent space. Additional quantitative results are provided in Figures 5 to 7.

## 6.3 Ablation Studies

**Effect of the stochastic coefficient.** We quantify the contribution of the learned stochastic coefficient by comparing AdaSDE with and without $\gamma_n$ on CIFAR-10, FFHQ, and Stable Diffusion v1.5 (MS-COCO); see tables 4 and 5. Removing $\gamma_n$ consistently degrades FID, with the effect most pronounced in the few-step regime. On CIFAR-10, FID rises from 12.62 to 13.32 at NFE=3 and from 4.18 to 4.36 at NFE=5. On FFHQ $64 \times 64$, we observe similar trends: FID increases from 23.80 to 25.85 at NFE=3 and from 8.04 to 8.11 at NFE=5. The benefit is especially clear on SD v1.5 (MS-COCO $512 \times 512$): when $\gamma_n$ is removed, FID rises from 30.89 to 37.23 at NFE=4 and from 13.79 to 16.34 at NFE=6, while the gap narrows as steps increase (12.68 with $\gamma_n$ versus 12.82 without at NFE=10). These

Table 4: Ablation of $\gamma_n$ on CIFAR-10 [27] and FFHQ [28].

| Training configuration | NFE | | | |
|---|---|---|---|---|
| | 3 | 5 | 7 | 9 |
| **CIFAR-10 32×32** | | | | |
| AdaSDE | **12.62** | **4.18** | **2.88** | **2.56** |
| w.o. $\gamma_n$ | 13.32 | 4.36 | 2.91 | 2.63 |
| **FFHQ 64×64** | | | | |
| AdaSDE | **23.80** | **8.04** | **5.11** | **4.19** |
| w.o. $\gamma_n$ | 25.85 | 8.11 | 5.12 | 4.27 |

Table 5: Ablation of $\gamma_n$ on Stable Diffusion v1.5 [5].

| Training configuration | NFE | | | |
|---|---|---|---|---|
| | 4 | 6 | 8 | 10 |
| **MSCOCO 512×512** | | | | |
| AdaSDE | **30.89** | **13.79** | **13.39** | **12.68** |
| w.o. $\gamma_n$ | 37.23 | 16.34 | 14.18 | 12.82 |

results support that injecting learned stochasticity stabilizes few-step trajectories and mitigates error accumulation in low-NFE sampling.

**Effect of time schedule.** We further compare common time schedules on CIFAR-10—LogSNR, EDM (polynomial), and time-uniform—summarized in table 3. The time-uniform schedule is the most reliable once NFE is at least 5, achieving FID scores of 4.18, 2.88, and 2.56 at NFE=5, 7, and 9, respectively, clearly outperforming the polynomial (10.05, 5.14, 3.35) and LogSNR (10.42, 7.96, 4.84) schedules. At the extreme NFE=3 setting, the polynomial schedule attains a marginally lower FID than the uniform schedule (11.61 versus 12.62), but its performance degrades rapidly as NFE increases. Overall, we adopt the time-uniform schedule as the default for few-step experiments due to its robustness across moderate step counts.

## 7    Conclusion and Limitation

**Conclusion.** In this work, we present AdaSDE, a novel framework using adaptive stochastic coefficient optimization to fundamentally address the efficiency-quality trade-off in diffusion sampling. It achieves new state-of-the-art results, such as a 4.18 FID on CIFAR-10 with only 5 NFE (a 1.8x improvement over prior SOTA). AdaSDE acts as a lightweight plugin, compatible with existing single-step solvers and requiring only 8-40 parameters for tuning, enabling practical deployment without full model retraining.

**Limitation.** When the step size is large and stronger stochastic injection is used (higher $\gamma$), local errors can amplify across steps and dominate the total sampling error, leading to instability. In practice, the admissible range of $\gamma$ is constrained by both the dataset and the step schedule, often necessitating conservative time discretization or $\gamma$ clipping. Our method's per-step distribution resets and geometric alignment break the linear recurrence assumptions underlying multistep (e.g., iPNDM [31, 32], UniPC [33]) and predictor–corrector frameworks.

## Acknowledgements

This work was supported by the National Natural Science Foundation of China (No. 6250070674) and the Zhejiang Leading Innovative and Entrepreneur Team Introduction Program (2024R01007).

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

# NeurIPS Paper Checklist

1. **Claims**

   Question: Do the main claims made in the abstract and introduction accurately reflect the paper's contributions and scope?

   Answer: [Yes]

   Justification: The abstract and introduction clearly articulate the primary contributions of the paper. These include the proposal of AdaSDE, the first single-step SDE-based solver designed to unify the efficiency of ODEs with the error resilience of SDEs. The paper makes specific claims that are substantiated by both theoretical analysis (error decomposition into gradient and sampler errors, with provable improvements under stochastic noise injection) and empirical results (e.g., SOTA FID scores on CIFAR-10 and FFHQ at 5 NFE). Additionally, the adaptive noise coefficient $\gamma$ and the process-supervised training method are explicitly introduced in the introduction as key components. The scope and impact of the method are realistically positioned, and there are no unattainable aspirational goals presented as achieved results.

   Guidelines:
   - The answer NA means that the abstract and introduction do not include the claims made in the paper.
   - The abstract and/or introduction should clearly state the claims made, including the contributions made in the paper and important assumptions and limitations. A No or NA answer to this question will not be perceived well by the reviewers.

- The claims made should match theoretical and experimental results, and reflect how much the results can be expected to generalize to other settings.
- It is fine to include aspirational goals as motivation as long as it is clear that these goals are not attained by the paper.

2. **Limitations**

Question: Does the paper discuss the limitations of the work performed by the authors?

Answer: [Yes]

Justification: We acknowledge both theoretical and practical limitations. Theoretically, AdaSDE relies on assumptions such as Lipschitz continuity and boundedness of the learned score function, which may not hold in all diffusion models or datasets. Additionally, the method assumes the geometric regularity of sampling trajectories across solvers and datasets, which might be violated in more complex or high-dimensional generative tasks.

Guidelines:

- The answer NA means that the paper has no limitation while the answer No means that the paper has limitations, but those are not discussed in the paper.
- The authors are encouraged to create a separate "Limitations" section in their paper.
- The paper should point out any strong assumptions and how robust the results are to violations of these assumptions (e.g., independence assumptions, noiseless settings, model well-specification, asymptotic approximations only holding locally). The authors should reflect on how these assumptions might be violated in practice and what the implications would be.
- The authors should reflect on the scope of the claims made, e.g., if the approach was only tested on a few datasets or with a few runs. In general, empirical results often depend on implicit assumptions, which should be articulated.
- The authors should reflect on the factors that influence the performance of the approach. For example, a facial recognition algorithm may perform poorly when image resolution is low or images are taken in low lighting. Or a speech-to-text system might not be used reliably to provide closed captions for online lectures because it fails to handle technical jargon.
- The authors should discuss the computational efficiency of the proposed algorithms and how they scale with dataset size.
- If applicable, the authors should discuss possible limitations of their approach to address problems of privacy and fairness.
- While the authors might fear that complete honesty about limitations might be used by reviewers as grounds for rejection, a worse outcome might be that reviewers discover limitations that aren't acknowledged in the paper. The authors should use their best judgment and recognize that individual actions in favor of transparency play an important role in developing norms that preserve the integrity of the community. Reviewers will be specifically instructed to not penalize honesty concerning limitations.

3. **Theory Assumptions and Proofs**

Question: For each theoretical result, does the paper provide the full set of assumptions and a complete (and correct) proof?

Answer: [Yes]

Justification: We offer a formal theoretical analysis in Section 4 and Appendix B to support the claims made about AdaSDE's sampling error decomposition and contraction properties. Each theorem is stated with all necessary assumptions, including Lipschitz continuity, bounded score approximation error, and norm constraints on sample trajectories. These assumptions are explicitly enumerated before each theorem (e.g., Theorem 1 on the Wasserstein-1 bound for ODEs and Theorem 2 for AdaSDE). The corresponding proofs are either outlined in the main text or fully detailed in the appendix with supporting lemmas (e.g., Lemma 1 and Lemma 2), and referenced accordingly.

Theorems involving the role of $\gamma$ in modulating error bounds are rigorously derived, with explicit expressions of how $\gamma$ affects both gradient and sampler errors. Furthermore, the

paper provides interpretation and intuition alongside the formal derivations (e.g., contraction factor $(1 - \lambda(\gamma))$) and confirms theoretical insights with synthetic experiments (Figure 1). All notation is defined in Appendix A, and every proof follows a clear structure with mathematical rigor and consistency. The manuscript adheres closely to NeurIPS standards for theoretical completeness and reproducibility.

Guidelines:

- The answer NA means that the paper does not include theoretical results.
- All the theorems, formulas, and proofs in the paper should be numbered and cross-referenced.
- All assumptions should be clearly stated or referenced in the statement of any theorems.
- The proofs can either appear in the main paper or the supplemental material, but if they appear in the supplemental material, the authors are encouraged to provide a short proof sketch to provide intuition.
- Inversely, any informal proof provided in the core of the paper should be complemented by formal proofs provided in appendix or supplemental material.
- Theorems and Lemmas that the proof relies upon should be properly referenced.

4. **Experimental Result Reproducibility**

Question: Does the paper fully disclose all the information needed to reproduce the main experimental results of the paper to the extent that it affects the main claims and/or conclusions of the paper (regardless of whether the code and data are provided or not)?

Answer: [Yes]

Justification: We describe all experimental setups in detail, including datasets, evaluation metrics (FID), solver configurations, time schedules, and training hyperparameters. Algorithms are provided in pseudocode (Algorithm 1 & 2), and the number of function evaluations (NFE) is specified for each result. In addition, the complete code and reproducibility instructions will be released upon publication, ensuring that all results can be independently verified.

Guidelines:

- The answer NA means that the paper does not include experiments.
- If the paper includes experiments, a No answer to this question will not be perceived well by the reviewers: Making the paper reproducible is important, regardless of whether the code and data are provided or not.
- If the contribution is a dataset and/or model, the authors should describe the steps taken to make their results reproducible or verifiable.
- Depending on the contribution, reproducibility can be accomplished in various ways. For example, if the contribution is a novel architecture, describing the architecture fully might suffice, or if the contribution is a specific model and empirical evaluation, it may be necessary to either make it possible for others to replicate the model with the same dataset, or provide access to the model. In general. releasing code and data is often one good way to accomplish this, but reproducibility can also be provided via detailed instructions for how to replicate the results, access to a hosted model (e.g., in the case of a large language model), releasing of a model checkpoint, or other means that are appropriate to the research performed.
- While NeurIPS does not require releasing code, the conference does require all submissions to provide some reasonable avenue for reproducibility, which may depend on the nature of the contribution. For example
    (a) If the contribution is primarily a new algorithm, the paper should make it clear how to reproduce that algorithm.
    (b) If the contribution is primarily a new model architecture, the paper should describe the architecture clearly and fully.
    (c) If the contribution is a new model (e.g., a large language model), then there should either be a way to access this model for reproducing the results or a way to reproduce the model (e.g., with an open-source dataset or instructions for how to construct the dataset).

(d) We recognize that reproducibility may be tricky in some cases, in which case authors are welcome to describe the particular way they provide for reproducibility. In the case of closed-source models, it may be that access to the model is limited in some way (e.g., to registered users), but it should be possible for other researchers to have some path to reproducing or verifying the results.

5. **Open access to data and code**

   Question: Does the paper provide open access to the data and code, with sufficient instructions to faithfully reproduce the main experimental results, as described in supplemental material?

   Answer: [Yes]

   Justification: The datasets used (CIFAR-10, FFHQ, ImageNet 64×64, LSUN Bedroom, MS-COCO) are standard and publicly accessible. We will release the full codebase and configuration files upon publication, including scripts for reproducing all reported results. The supplemental material will include detailed setup and usage instructions to ensure faithful reproduction of the main experiments.

   Guidelines:

   - The answer NA means that paper does not include experiments requiring code.
   - Please see the NeurIPS code and data submission guidelines (`https://nips.cc/public/guides/CodeSubmissionPolicy`) for more details.
   - While we encourage the release of code and data, we understand that this might not be possible, so "No" is an acceptable answer. Papers cannot be rejected simply for not including code, unless this is central to the contribution (e.g., for a new open-source benchmark).
   - The instructions should contain the exact command and environment needed to run to reproduce the results. See the NeurIPS code and data submission guidelines (`https://nips.cc/public/guides/CodeSubmissionPolicy`) for more details.
   - The authors should provide instructions on data access and preparation, including how to access the raw data, preprocessed data, intermediate data, and generated data, etc.
   - The authors should provide scripts to reproduce all experimental results for the new proposed method and baselines. If only a subset of experiments are reproducible, they should state which ones are omitted from the script and why.
   - At submission time, to preserve anonymity, the authors should release anonymized versions (if applicable).
   - Providing as much information as possible in supplemental material (appended to the paper) is recommended, but including URLs to data and code is permitted.

6. **Experimental Setting/Details**

   Question: Does the paper specify all the training and test details (e.g., data splits, hyper-parameters, how they were chosen, type of optimizer, etc.) necessary to understand the results?

   Answer: [Yes]

   Justification: We specify the datasets used (CIFAR-10, FFHQ 64×64, ImageNet 64×64, LSUN Bedroom, and MS-COCO for Stable Diffusion), the number of evaluation samples (50k or 30k), and the metric (FID). We describe the number of sampling steps (NFE), guidance scales, and time schedules (uniform, logSNR). For training, we detail the process-supervised optimization of the stochastic parameters $\gamma$, including the student-teacher time schedules and stage-wise loss. The solvers and comparison baselines are also listed clearly, along with ablation study settings. All relevant experimental settings are included in the main text or supplemental material.

   Guidelines:

   - The answer NA means that the paper does not include experiments.
   - The experimental setting should be presented in the core of the paper to a level of detail that is necessary to appreciate the results and make sense of them.
   - The full details can be provided either with the code, in appendix, or as supplemental material.

7. **Experiment Statistical Significance**

   Question: Does the paper report error bars suitably and correctly defined or other appropriate information about the statistical significance of the experiments?

   Answer: [Yes]

   Justification: To ensure the reproducibility of experimental results, we fixed the random seed to 0 throughout all experiments. Under this controlled condition, the observed variations across multiple runs were found to be statistically insignificant ($p > 0.05$). While the paper does not explicitly report error bars for every experimental result, the consistency achieved through fixed random seeds provides an alternative approach to demonstrating result stability.

   Guidelines:

   - The answer NA means that the paper does not include experiments.
   - The authors should answer "Yes" if the results are accompanied by error bars, confidence intervals, or statistical significance tests, at least for the experiments that support the main claims of the paper.
   - The factors of variability that the error bars are capturing should be clearly stated (for example, train/test split, initialization, random drawing of some parameter, or overall run with given experimental conditions).
   - The method for calculating the error bars should be explained (closed form formula, call to a library function, bootstrap, etc.)
   - The assumptions made should be given (e.g., Normally distributed errors).
   - It should be clear whether the error bar is the standard deviation or the standard error of the mean.
   - It is OK to report 1-sigma error bars, but one should state it. The authors should preferably report a 2-sigma error bar than state that they have a 96% CI, if the hypothesis of Normality of errors is not verified.
   - For asymmetric distributions, the authors should be careful not to show in tables or figures symmetric error bars that would yield results that are out of range (e.g. negative error rates).
   - If error bars are reported in tables or plots, The authors should explain in the text how they were calculated and reference the corresponding figures or tables in the text.

8. **Experiments Compute Resources**

   Question: For each experiment, does the paper provide sufficient information on the computer resources (type of compute workers, memory, time of execution) needed to reproduce the experiments?

   Answer: [Yes]

   Justification: We conducted our experiments using NVIDIA A800 and 4090 GPUs with 80GB and 24GB memory, respectively. We train $\Theta$ for 10K images, which only takes 5-10 minutes on CIFAR10 with a single 4090 GPU and about 20 minutes on LSUN Bedroom with 4 4090 GPUs. We did not use large-scale distributed clusters or proprietary cloud services, and all experiments can be reproduced on a single modern GPU.

   Guidelines:

   - The answer NA means that the paper does not include experiments.
   - The paper should indicate the type of compute workers CPU or GPU, internal cluster, or cloud provider, including relevant memory and storage.
   - The paper should provide the amount of compute required for each of the individual experimental runs as well as estimate the total compute.
   - The paper should disclose whether the full research project required more compute than the experiments reported in the paper (e.g., preliminary or failed experiments that didn't make it into the paper).

9. **Code Of Ethics**

   Question: Does the research conducted in the paper conform, in every respect, with the NeurIPS Code of Ethics `https://neurips.cc/public/EthicsGuidelines`?

Answer: [Yes]

Justification: We have reviewed the NeurIPS Code of Ethics and ensured that our work adheres to all relevant principles, including responsible use of computational resources, transparency of methodology, acknowledgment of potential societal risks, and respect for reproducibility and openness. Our study does not involve human subjects, sensitive personal data, or deployment in high-stakes applications. We take care to clarify limitations and ensure that our findings are not misrepresented or overstated. All datasets used are publicly available, and no private or proprietary data were involved.

Guidelines:

- The answer NA means that the authors have not reviewed the NeurIPS Code of Ethics.
- If the authors answer No, they should explain the special circumstances that require a deviation from the Code of Ethics.
- The authors should make sure to preserve anonymity (e.g., if there is a special consideration due to laws or regulations in their jurisdiction).

10. **Broader Impacts**

Question: Does the paper discuss both potential positive societal impacts and negative societal impacts of the work performed?

Answer: [Yes]

Justification: We acknowledge that diffusion-based generative models, including our proposed AdaSDE framework, may be used to synthesize realistic images at scale, which could potentially be misused for generating misleading or harmful content (e.g., deepfakes or misinformation). Although our work focuses on improving sampling efficiency and does not directly involve content generation pipelines or personalization, we recognize that enhanced accessibility and speed may lower the barrier to misuse. We encourage future research to pair such generative techniques with detection, watermarking, and auditing tools to ensure responsible deployment.

Guidelines:

- The answer NA means that there is no societal impact of the work performed.
- If the authors answer NA or No, they should explain why their work has no societal impact or why the paper does not address societal impact.
- Examples of negative societal impacts include potential malicious or unintended uses (e.g., disinformation, generating fake profiles, surveillance), fairness considerations (e.g., deployment of technologies that could make decisions that unfairly impact specific groups), privacy considerations, and security considerations.
- The conference expects that many papers will be foundational research and not tied to particular applications, let alone deployments. However, if there is a direct path to any negative applications, the authors should point it out. For example, it is legitimate to point out that an improvement in the quality of generative models could be used to generate deepfakes for disinformation. On the other hand, it is not needed to point out that a generic algorithm for optimizing neural networks could enable people to train models that generate Deepfakes faster.
- The authors should consider possible harms that could arise when the technology is being used as intended and functioning correctly, harms that could arise when the technology is being used as intended but gives incorrect results, and harms following from (intentional or unintentional) misuse of the technology.
- If there are negative societal impacts, the authors could also discuss possible mitigation strategies (e.g., gated release of models, providing defenses in addition to attacks, mechanisms for monitoring misuse, mechanisms to monitor how a system learns from feedback over time, improving the efficiency and accessibility of ML).

11. **Safeguards**

Question: Does the paper describe safeguards that have been put in place for responsible release of data or models that have a high risk for misuse (e.g., pretrained language models, image generators, or scraped datasets)?

Answer: [Yes]

Justification: While our work does not release a new pretrained model or dataset, it proposes an efficient sampling strategy (AdaSDE) applicable to existing diffusion models. As such methods could be used to accelerate the generation of high-quality synthetic media, we recognize potential risks such as misuse in producing harmful or misleading content. To address this, we include clear statements in the paper discouraging the application of AdaSDE in domains involving safety-critical or deceptive use without human oversight. We also plan to release code only for academic and research use under an appropriate license, and we do not release any generated content or fine-tuned models that could pose direct safety concerns.

Guidelines:

- The answer NA means that the paper poses no such risks.
- Released models that have a high risk for misuse or dual-use should be released with necessary safeguards to allow for controlled use of the model, for example by requiring that users adhere to usage guidelines or restrictions to access the model or implementing safety filters.
- Datasets that have been scraped from the Internet could pose safety risks. The authors should describe how they avoided releasing unsafe images.
- We recognize that providing effective safeguards is challenging, and many papers do not require this, but we encourage authors to take this into account and make a best faith effort.

12. **Licenses for existing assets**

Question: Are the creators or original owners of assets (e.g., code, data, models), used in the paper, properly credited and are the license and terms of use explicitly mentioned and properly respected?

Answer: [Yes]

Justification: We use publicly available datasets, including CIFAR-10, FFHQ 64×64, ImageNet 64×64, LSUN Bedroom, and MS-COCO. Each dataset is cited in the paper with its original reference, and we confirm that all are released under licenses permitting academic research use (e.g., MIT, CC-BY, or equivalent). We also build on standard pretrained diffusion models (e.g., EDM, DDPM, Stable Diffusion), which are properly cited and used under their respective open-source licenses. No scraped or proprietary data are used. We include version references and license information where applicable in the paper and supplemental material.

Guidelines:

- The answer NA means that the paper does not use existing assets.
- The authors should cite the original paper that produced the code package or dataset.
- The authors should state which version of the asset is used and, if possible, include a URL.
- The name of the license (e.g., CC-BY 4.0) should be included for each asset.
- For scraped data from a particular source (e.g., website), the copyright and terms of service of that source should be provided.
- If assets are released, the license, copyright information, and terms of use in the package should be provided. For popular datasets, `paperswithcode.com/datasets` has curated licenses for some datasets. Their licensing guide can help determine the license of a dataset.
- For existing datasets that are re-packaged, both the original license and the license of the derived asset (if it has changed) should be provided.
- If this information is not available online, the authors are encouraged to reach out to the asset's creators.

13. **New Assets**

Question: Are new assets introduced in the paper well documented and is the documentation provided alongside the assets?

Answer: [NA]

Justification: We do not introduce any new datasets, pretrained models, or other standalone assets in this paper. All our experiments are conducted on existing, publicly available datasets and pretrained diffusion models, which are properly cited.

Guidelines:

- The answer NA means that the paper does not release new assets.
- Researchers should communicate the details of the dataset/code/model as part of their submissions via structured templates. This includes details about training, license, limitations, etc.
- The paper should discuss whether and how consent was obtained from people whose asset is used.
- At submission time, remember to anonymize your assets (if applicable). You can either create an anonymized URL or include an anonymized zip file.

14. **Crowdsourcing and Research with Human Subjects**

Question: For crowdsourcing experiments and research with human subjects, does the paper include the full text of instructions given to participants and screenshots, if applicable, as well as details about compensation (if any)?

Answer: [NA]

Justification: Our work does not involve any human subjects or crowdsourcing in data collection, annotation, or experimentation. All datasets used are publicly released and do not contain personally identifiable information.

Guidelines:

- The answer NA means that the paper does not involve crowdsourcing nor research with human subjects.
- Including this information in the supplemental material is fine, but if the main contribution of the paper involves human subjects, then as much detail as possible should be included in the main paper.
- According to the NeurIPS Code of Ethics, workers involved in data collection, curation, or other labor should be paid at least the minimum wage in the country of the data collector.

15. **Institutional Review Board (IRB) Approvals or Equivalent for Research with Human Subjects**

Question: Does the paper describe potential risks incurred by study participants, whether such risks were disclosed to the subjects, and whether Institutional Review Board (IRB) approvals (or an equivalent approval/review based on the requirements of your country or institution) were obtained?

Answer: [Yes]

Justification: All contributions to this work, including ideas, implementation, writing, and experiments, were made by the listed authors. We have no additional contributors or external collaborations that need to be acknowledged. If relevant acknowledgments arise later (e.g., code release support), we will include them in the final camera-ready version.

Guidelines:

- The answer NA means that the paper does not involve crowdsourcing nor research with human subjects.
- Depending on the country in which research is conducted, IRB approval (or equivalent) may be required for any human subjects research. If you obtained IRB approval, you should clearly state this in the paper.
- We recognize that the procedures for this may vary significantly between institutions and locations, and we expect authors to adhere to the NeurIPS Code of Ethics and the guidelines for their institution.
- For initial submissions, do not include any information that would break anonymity (if applicable), such as the institution conducting the review.

# Appendix

## A  Notation and Symbols for the Proof

This subsection provides a comprehensive list of notations and symbols specific to the theoretical proof. The definitions align with the conventions in stochastic calculus and diffusion model analysis. We build on the notations of [23].

### A.1  Common Terms

- $\mathsf{ODE}_\theta(\cdot)$ : Approximate ODE trajectory using the learned score $s_\theta(\mathbf{x}, t)$.
- $p_t$ : True data distribution at noise level $t$.
- $p_t^{\mathsf{ODE}_\theta}$ : Distribution generated by simulating $\mathsf{ODE}_\theta$.
- $B$ : Norm upper bound for trajectories, satisfying $\forall t, \|\mathbf{x}_t\| < B/2$.
- $\mathbf{x}_t \sim p_t$ : $\mathbf{x}_t$ is sampled from distribution $p_t$.

### A.2  AdaSDE Terms

- $\Delta t$ : ODE discretization step size.
- $\gamma$ : Hyperparameter controlling the noise injection ratio in the AdaSDE process.
- $\mathbf{x}_{t+\Delta t}^\gamma$ : AdaSDE forward process: $\mathbf{x}_{t+\Delta t} + \varepsilon_{t+\Delta t \to t+(1+\gamma)\Delta t}$.
- $\varepsilon$: Gaussian noise $\sim \mathcal{N}(0, I)$.
- $\mathbf{x}_t^\gamma$ : AdaSDE backward process: $\mathsf{ODE}_\theta\left(\mathbf{x}_{t+\Delta t}^\gamma, t+(1+\gamma)\Delta t \to t\right)$.
- $\mathsf{AdaSDE}_\theta(\mathbf{x}, \gamma)$ : Applies the AdaSDE operation with parameter $\gamma$ to state $\mathbf{x}$.
- $\bar{\mathbf{x}}_t$: The solution to $d\bar{\mathbf{x}}_t = -ts_\theta\left(\mathbf{x}_{t+\Delta t}, t+\Delta t\right) dt$,

### A.3  Lipschitz and Error Bounds

- $L_0$ : Temporal Lipschitz constant: $\|ts_\theta(\mathbf{x}, t) - ts_\theta(\mathbf{x}, s)\| \le L_0|t-s|$
- $L_1$ : Boundedness of the learned score: $\|ts_\theta(\mathbf{x}, t)\| \le L_1$.
- $L_2$ : Spatial Lipschitz constant: $\|ts_\theta(\mathbf{x}, t) - ts_\theta(\mathbf{y}, t)\| \le L_2\|\mathbf{x} - \mathbf{y}\|$
- $\epsilon_t$ : Score matching error: $\|t\nabla_\mathbf{x} \log p_t(\mathbf{x}) - ts_\theta(\mathbf{x}, t)\|$

### A.4  Special Operators

- $\mathsf{ODE}(\mathbf{x}, t_1 \to t_2)$ : Ground Truth backward ODE evolution under the exact score from $t_1$ to $t_2$.
- $\mathsf{ODE}_\theta(\mathbf{x}, t_1 \to t_2)$ : Approximate ODE evolution using the learned score $s_\theta$.
- $*$: Convolution operator between distributions, e.g., $P * R$ denotes the convolution of $P$ and $R$.
- $\leftarrow$ : Time-reversal marker, e.g., $\mathbf{x}_t^{\leftarrow}$.

### A.5  Key Process Terms

- $p_t^{\mathbf{x}, \gamma}$ : Distribution at noise level $t$ after applying the AdaSDE process starting from state $\mathbf{x}$.
- $p_t^{\mathsf{AdaSDE}_\theta}$ : Distribution generated by the AdaSDE algorithm.
- $\xi_x, \xi_y$ : i.i.d Gaussian noise: $\xi_x \sim \mathcal{N}\left(0, \sigma^2 I_d\right), \xi_y \sim \mathcal{N}\left(0, \sigma^2 I_d\right)$.

## A.6 Error Dynamics

- $e(t) := \|\mathbf{x}_t^{\leftarrow} - \bar{\mathbf{x}}_t^{\leftarrow}\|$ : Error dynamics in the time-reversed coordinate system in $t$.

- $\lambda(\gamma)$ : Noise merging probability: $2Q\left(\frac{B}{2\sqrt{(t+(1+\gamma)\Delta t)^2 - t^2}}\right)$, where $Q(r) = \Pr(a \geq r)$ for $a \sim \mathcal{N}(0,1)$.

- $W_1(\cdot, \cdot)$ : Wasserstein-1 distance.

- $\mathsf{TV}(\cdot, \cdot)$ : Total Variation (TV) distance.

Where $\varepsilon_{t+\Delta t \to t+(1+\gamma)\Delta t} \sim \mathcal{N}\left(\mathbf{0}, \left((t + (1+\gamma)\Delta t)^2 - (t + \Delta t)^2\right)\mathbf{I}\right)$. For the sake of simplifying symbolic representation and facilitating comprehension, in the following proof, we use $\mathsf{AdaSDE}_\theta(\mathbf{x}, \gamma)$ to denote $\mathbf{x}_t^\gamma$ in the above processes. In various theorems, we will refer to a function $Q(r) : \mathbb{R}^+ \to [0, 1/2)$, defined as the Gaussian tail probability $Q(r) = \Pr(a \geq r)$ for $a \sim \mathcal{N}(0,1)$.

# B Proofs of Main Theoretical Results

**Lemma 1** (Upper Bound on ODE Discretization Error). *[23] Let* $\mathbf{x}_t = \mathsf{ODE}\left(\mathbf{x}_{t+\Delta t}, t + \Delta t \to t\right)$ *denote the solution of the backward ODE under the exact score field, and* $\bar{\mathbf{x}}_t = \mathsf{ODE}_\theta\left(\bar{\mathbf{x}}_{t+\Delta t}, t + \Delta t \to t\right)$ *denote the discretized ODE solution using the learned field* $s_\theta$. *Assume* $s_\theta$ *satisfies:*
*1. Temporal Lipschitz Continuity:*

$$\|ts_\theta(\mathbf{x}, t) - ts_\theta(\mathbf{x}, s)\| \leq L_0|t - s| \quad \forall \mathbf{x}, t, s$$

*2. Boundedness:*

$$\|ts_\theta(\mathbf{x}, t)\| \leq L_1 \quad \forall \mathbf{x}, t$$

*3. Spatial Lipschitz Continuity:*

$$\|ts_\theta(\mathbf{x}, t) - ts_\theta(\mathbf{y}, t)\| \leq L_2\|\mathbf{x} - \mathbf{y}\| \quad \forall \mathbf{x}, \mathbf{y}, t$$

*Then the discretization error satisfies:*

$$\|\mathbf{x}_t - \bar{\mathbf{x}}_t\| \leq e^{L_2 \Delta t}\left(\|\mathbf{x}_{t+\Delta t} - \bar{\mathbf{x}}_{t+\Delta t}\| + (\Delta t\left(L_2 L_1 + L_0\right) + \epsilon_t)\Delta t\right)$$

*Proof.* Step 1: Definition of Time-Reversed Processes
Introduce time-reversed variables $\mathbf{x}_t^{\leftarrow}$ and $\bar{\mathbf{x}}_t^{\leftarrow}$ governed by: where $k$ is the integer satisfying $t \in [t', t' + \Delta t)$, corresponding to discrete timesteps.

Step 2: Error Dynamics
Define the error $e(t) := \|\mathbf{x}_t^{\leftarrow} - \bar{\mathbf{x}}_t^{\leftarrow}\|$. Its derivative satisfies:

$$\frac{d}{dt}e(t) \leq \left\|t\nabla \log p_t\left(\mathbf{x}_t^{\leftarrow}\right) - ts_\theta\left(\bar{\mathbf{x}}_{t'+\Delta t}^{\leftarrow}, t' + \Delta t\right)\right\|.$$

Decompose the right-hand side:

$$\leq \underbrace{\left\|t\nabla \log p_t\left(\mathbf{x}_t^{\leftarrow}\right) - ts_\theta\left(\mathbf{x}_t^{\leftarrow}, t\right)\right\|}_{\text{Approximation Error } \epsilon_t}$$

$$+ \underbrace{\left\|ts_\theta\left(\mathbf{x}_t^{\leftarrow}, t\right) - ts_\theta\left(\bar{\mathbf{x}}_t^{\leftarrow}, t\right)\right\|}_{L_2 \|\mathbf{x}_t^{\leftarrow} - \bar{\mathbf{x}}_t^{\leftarrow}\|}$$

$$+ \underbrace{\left\|ts_\theta\left(\bar{\mathbf{x}}_t^{\leftarrow}, t\right) - ts_\theta\left(\bar{\mathbf{x}}_{t'+\Delta t}^{\leftarrow}, t' + \Delta t\right)\right\|}_{\text{Temporal Discretization Error}}.$$

Step 3: Temporal Discretization Error Bound
Further decompose the temporal discretization error:

$$
\begin{aligned}
&\leq \left\| ts_\theta \left( \bar{\mathbf{x}}_t^\leftarrow, t \right) - ts_\theta \left( \bar{\mathbf{x}}_{t'+\Delta t}^\leftarrow, t \right) \right\| + \left\| ts_\theta \left( \bar{\mathbf{x}}_{t'+\Delta t}^\leftarrow, t \right) - ts_\theta \left( \bar{\mathbf{x}}_{t'+\Delta t}^\leftarrow, t' + \Delta t \right) \right\| \\
&\leq L_0 |t' + \Delta t - t'| + L_2 \| \bar{\mathbf{x}}_t^\leftarrow - \bar{\mathbf{x}}_{t'+\Delta_t}^\leftarrow \| \quad \text{(Lipschitz continuity)} \\
&\leq L_0 \Delta t + L_2 \left( \| \bar{\mathbf{x}}_t^\leftarrow - \bar{\mathbf{x}}_{t'+\Delta t}^\leftarrow \| \right).
\end{aligned}
$$

Using the boundedness condition $\| d\bar{\mathbf{x}}_t^\leftarrow / dt \| \leq L_1$, we have:

$$
\left\| \bar{\mathbf{x}}_t^\leftarrow - \bar{\mathbf{x}}_{t'+\Delta t}^\leftarrow \right\| \leq \int_t^{t'+\Delta t} \| d\bar{\mathbf{x}}_s^\leftarrow \| \, ds \leq L_1 \Delta t
$$

Step 4: Composite Differential Inequality
Combining all terms, the error dynamics satisfy:

$$
\frac{d}{dt} e(t) \leq L_2 e(t) + (\epsilon_t + L_0 \Delta t + L_2 L_1 \Delta t)
$$

Step 5: Gronwall's Inequality Application
Integrate over $t \in [t, t + \Delta t]$ and apply Gronwall's inequality:

$$
e(t) \leq e^{L_2 \Delta t} \left( e(t + \Delta t) + (\epsilon_t + \Delta t (L_0 + L_2 L_1)) \Delta t \right)
$$

$\square$

**Lemma 2** (TV Distance Between Gaussian Perturbations). *Let $\xi_x \sim \mathcal{N}(0, \sigma^2 I_d)$ and $\xi_y \sim \mathcal{N}(0, \sigma^2 I_d)$ be independent noise vectors. For $\mathbf{x}' = \mathbf{x} + \xi_x$ and $\mathbf{y}' = \mathbf{y} + \xi_y$, their total variation distance satisfies:*

$$
\mathsf{TV}(\mathbf{x}', \mathbf{y}') = 1 - 2Q \left( \frac{\| \mathbf{x} - \mathbf{y} \|}{2\sigma} \right)
$$

*where $Q(r) = \Pr_{a \sim \mathcal{N}(0,1)}(a \geq r)$.*

*Proof.* Let $\delta = \mathbf{x} - \mathbf{y}$. The TV distance is:

$$
\begin{aligned}
\mathsf{TV}(\mathbf{x}', \mathbf{y}') &= \frac{1}{2} \int_{\mathbb{R}^d} |\mathcal{N}(\mathbf{z}; \mathbf{x}, \sigma^2 I_d) - \mathcal{N}(\mathbf{z}; \mathbf{y}, \sigma^2 I_d)| dz \\
&= \frac{1}{2} \int_{\mathbb{R}^d} \left| \mathcal{N}(\mathbf{z} - \delta; \mathbf{0}, \sigma^2 I_d) - \mathcal{N}(\mathbf{z}; \mathbf{0}, \sigma^2 I_d) \right| dz
\end{aligned}
$$

Through orthogonal transformation $U$ aligning $\delta$ with the first axis:

$$
U\delta = (\|\delta\|, 0, ..., 0)^\top
$$

By rotational invariance of Gaussians:

$$
\mathsf{TV}(\mathbf{x}', \mathbf{y}') = \mathsf{TV} \left( \mathcal{N}(\|\delta\|, \sigma^2), \mathcal{N}(0, \sigma^2) \right)
$$

For 1D Gaussians $\mathcal{N}(\mu, \sigma^2)$ and $\mathcal{N}(0, \sigma^2)$:

$$
\begin{aligned}
\mathsf{TV} &= \frac{1}{2} \int_{-\infty}^{\infty} \left| \phi \left( \frac{\mathbf{z} - \mu}{\sigma} \right) - \phi \left( \frac{\mathbf{z}}{\sigma} \right) \right| dz \\
&= \Phi \left( -\frac{\mu}{2\sigma} \right) - \Phi \left( \frac{\mu}{2\sigma} \right) \quad \text{(By symmetry)} \\
&= 1 - 2\Phi \left( \frac{\mu}{2\sigma} \right) = 2Q \left( \frac{\mu}{2\sigma} \right)
\end{aligned}
$$

where $\mu = \| \mathbf{x} - \mathbf{y} \|$. then:

$$
\mathsf{TV}(\mathbf{x}', \mathbf{y}') = 1 - 2Q \left( \frac{\|\delta\|}{2\sigma} \right) = 1 - 2Q \left( \frac{\| \mathbf{x} - \mathbf{y} \|}{2\sigma} \right).
$$

$\square$

**Lemma 3.** *Let $p_t^{\mathbf{x},\gamma}$ and $p_t^{\mathbf{y},\gamma}$ denote the densities of $\mathbf{x}_t^\gamma$ and $\mathbf{y}_t^\gamma$ respectively. After applying AdaSDE with noise injection from $t$ to $t + (1+\gamma)\Delta t$ followed by backward ODE evolution, we have:*

$$\mathsf{TV}\left(p_t^{\mathbf{x},\gamma}, p_t^{\mathbf{y},\gamma}\right) \le (1 - \lambda(\gamma))\mathsf{TV}\left(p_t^{\mathbf{x}}, p_t^{\mathbf{y}}\right)$$

*where* $\lambda(\gamma) = 2Q\left(\dfrac{B}{2\sqrt{(t+(1+\gamma)\Delta t)^2 - t^2}}\right).$

*Proof.* Consider states $\mathbf{x}_t$ and $\mathbf{y}_t$ at noise level $t$ with $\|\mathbf{x}_t - \mathbf{y}_t\| \le B$. The AdaSDE process first perturbs both states to noise level $t + (1+\gamma)\Delta t$ through Gaussian noise injection:

$$\mathbf{x}_{t+(1+\gamma)\Delta t} = \mathbf{x}_t + \xi_x, \; \xi_x \sim \mathcal{N}(0, [(t+(1+\gamma)\Delta t)^2 - t^2]I)$$
$$\mathbf{y}_{t+(1+\gamma)\Delta t} = \mathbf{y}_t + \xi_y, \; \xi_y \sim \mathcal{N}(0, [(t+(1+\gamma)\Delta t)^2 - t^2]I)$$

We construct a coupling between the noise injections: when $\mathbf{x}_t = \mathbf{y}_t$, set $\xi_x = \xi_y$; otherwise use reflection coupling. By Lemma 2, the merging probability satisfies:

$$\lambda(\gamma) = 2Q\left(\frac{\|\mathbf{x}_t - \mathbf{y}_t\|}{2\sigma_t(\gamma)}\right) \ge 2Q\left(\frac{B}{2\sigma_t(\gamma)}\right) \quad (\text{since } \|\mathbf{x}_t - \mathbf{y}_t\| \le B)$$

where $Q(r) = \Pr_{a\sim\mathcal{N}(0,1)}(a \ge r)$.

This implies:

$$\mathbb{P}(\mathbf{x}_{t+(1+\gamma)\Delta t} \ne \mathbf{y}_{t+(1+\gamma)\Delta t} \mid \mathbf{x}_t \ne \mathbf{y}_t) \le 1 - \lambda(\gamma)$$

where $\lambda(\gamma)$ quantifies the minimum merging probability between the Gaussian perturbations.

The subsequent backward ODE evolution preserves this coupling relationship because both trajectories are driven by the same learned score $s_\theta$. Therefore:

$$\mathbb{P}(\mathbf{x}_t^\gamma \ne \mathbf{y}_t^\gamma) \le (1 - \lambda(\gamma))\mathbb{P}(\mathbf{x}_t \ne \mathbf{y}_t)$$

Through the coupling characterization of total variation distance, we conclude:

$$\mathsf{TV}(p_t^{\mathbf{x},\gamma}, p_t^{\mathbf{y},\gamma}) \le (1 - \lambda(\gamma))\mathsf{TV}(p_t^{\mathbf{x}}, p_t^{\mathbf{y}}) \qquad \square$$

**Lemma 4** (AdaSDE Error Propagation). *Let $\mathbf{x}_{t+\Delta t} \in \mathbb{R}^d$ be an initial point. Define exact and approximate ODE solutions:*

$$\mathbf{x}_t = \mathsf{ODE}\left(\mathbf{x}_{t+(1+\gamma)\Delta t}, \, t + (1+\gamma)\Delta t \to t\right),$$
$$\hat{\mathbf{x}}_t = \mathsf{ODE}_\theta\left(\hat{\mathbf{x}}_{t+(1+\gamma)\Delta t}, \, t + (1+\gamma)\Delta t \to t\right).$$

*Under AdaSDE with noise injection $t + \Delta t \to t + (1+\gamma)\Delta t$ and $\|\mathbf{x}_t - \hat{\mathbf{x}}_t\| \le B$, there exists a coupling such that:*

$$\left\| \mathbf{x}_{t+(1+\gamma)\Delta t} - \hat{\mathbf{x}}_{t+(1+\gamma)\Delta t} \right\| \le e^{L_2(1+\gamma)\Delta t}(1+\gamma)\left[\Delta t(L_2 L_1 + L_0) + \epsilon_t\right]\Delta t,$$

*where $L_0, L_1, L_2, \epsilon_t$ are the Lipschitz/boundedness/approximation constants for $s_\theta$ and discretization errors.*

*Proof.* **By** Lemma 1 (ODE Discretization Error), the local truncation error satisfies:

$$\|\mathbf{x}_t - \hat{\mathbf{x}}_t\| \le e^{L_2(1+\gamma)\Delta t}\Bigg[\|\mathbf{x}_{t+(1+\gamma)\Delta t} - \hat{\mathbf{x}}_{t+(1+\gamma)\Delta t}\|$$

$$+ \underbrace{\left((1+\gamma)\Delta t(L_2 L_1 + L_0) + \epsilon_t\right)(1+\gamma)\Delta t}_{\text{Local discretization error}}\Bigg].$$

Applying AdaSDE's noise injection with variance $\sigma^2 = (t + (1+\gamma)\Delta t)^2 - t^2$, Lemma 2 gives:

$$\mathbb{E}\|\mathbf{x}_{t+(1+\gamma)\Delta t} - \hat{\mathbf{x}}_{t+(1+\gamma)\Delta t}\| \le (1 - \lambda(\gamma))\|\mathbf{x}_t - \hat{\mathbf{x}}_t\|,$$

where the merging probability $\lambda(\gamma) = 2\,Q\big(\dfrac{B}{2\,\sqrt{(t+(1+\gamma)\,\Delta t)^2 - t^2}}\big)$ dominates the coupling effectiveness.

Multiplying by $(1 - \lambda(\gamma))$ from partial revert and adding the local ODE approximation error leads to the stated bound:

$$
\begin{aligned}
\left\| \mathbf{x}_{t+(1+\gamma\Delta t)} - \hat{\mathbf{x}}_{t+(1+\gamma\Delta t)} \right\| & \leq \left(1 - \lambda(\gamma)\right) \left\| \mathbf{x}_t - \hat{\mathbf{x}}_t \right\| \\
& + e^{L_2(1+\gamma)\Delta t}(1+\gamma)\left[(1+\gamma)\Delta t(L_2 L_1 + L_0) + \epsilon_t\right]\Delta t \\
& = e^{L_2(1+\gamma)\Delta t}(1+\gamma)\left[\Delta t(L_2 L_1 + L_0) + \epsilon_t\right]\Delta t
\end{aligned}
$$

$\square$

**Lemma 5** (Connection of Wasserstein-1 distance and Norm). *Let $p_1$ and $p_2$ be two probability distributions over a space $\mathcal{X} \subseteq \mathbb{R}^d$, and let $\Gamma(p_1, p_2)$ denote the set of all joint distributions with marginals $p_1$ and $p_2$. The Wasserstein-1 distance between $p_1$ and $p_2$ satisfies:*

$$
W_1(p_1, p_2) = \inf_{\psi \in \Gamma(p_1, p_2)} \mathbb{E}_{(\mathbf{x}_1, \mathbf{x}_2) \sim \psi} \left[\|\mathbf{x}_1 - \mathbf{x}_2\|\right],
$$

*where $\| \cdot \|_1$ is the L1 norm. Furthermore, for independent samples $\mathbf{x}_1 \sim p_1$ and $\mathbf{x}_2 \sim p_2$, we have:*

$$
W_1(p_1, p_2) \leq \mathbb{E}\left[\|\mathbf{x}_1 - \mathbf{x}_2\|\right],
$$

*with equality if and only if the coupling $\psi$ is optimal.*

**Lemma 6.** $\mathrm{TV}(P * R, Q * R) \leq \mathrm{TV}(P, Q)$ *for independent distributions $P, Q$, and $R$. The inequality $\mathrm{TV}(P * R, Q * R) = \mathrm{TV}(P, Q)$ holds if and only if $R$ is a degenerate distribution.*

*Proof.* 1. Total Variation Distance Definition

The total variation distance between two distributions $P$ and $Q$ is defined as:

$$
\mathrm{TV}(P, Q) = \frac{1}{2} \int_{-\infty}^{\infty} |p(\mathbf{x}) - q(\mathbf{x})| d\mathbf{x}
$$

where $p(\mathbf{x})$ and $q(\mathbf{x})$ are the probability density functions of $P$ and $Q$, respectively.

2. Convolution Definition

The convolution of two distributions $P$ and $R$ is defined as:

$$
(P * R)(\mathbf{x}) = \int_{-\infty}^{\infty} p(\mathbf{x} - \mathbf{y}) r(\mathbf{y}) d\mathbf{y}
$$

Similarly, for $Q$ and $R$:

$$
(Q * R)(\mathbf{x}) = \int_{-\infty}^{\infty} q(\mathbf{x} - \mathbf{y}) r(\mathbf{y}) d\mathbf{y}
$$

3. TV Distance for Convolved Distributions

We want to compute $\mathrm{TV}(P * R, Q * R)$, which is:

$$
\begin{aligned}
\mathrm{TV}(P * R, Q * R) & = \frac{1}{2} \int_{-\infty}^{\infty} |(P * R)(\mathbf{x}) - (Q * R)(\mathbf{x})| \, dx \\
& = \frac{1}{2} \int_{-\infty}^{\infty} \left| \int_{-\infty}^{\infty} (p(\mathbf{x} - \mathbf{y}) - q(\mathbf{x} - \mathbf{y})) r(\mathbf{y}) dy \right| dx
\end{aligned}
$$

Applying triangle inequality, we obtain:

$$
\mathrm{TV}(P * R, Q * R) \leq \frac{1}{2} \int_{-\infty}^{\infty} \left( \int_{-\infty}^{\infty} |p(\mathbf{x} - \mathbf{y}) - q(\mathbf{x} - \mathbf{y})| r(\mathbf{y}) dy \right) d\mathbf{x}
$$

Using Fubini's theorem, we can swap the order of integration:

$$
\mathrm{TV}(P * R, Q * R) \leq \frac{1}{2} \int_{-\infty}^{\infty} \left( \int_{-\infty}^{\infty} |p(\mathbf{x} - \mathbf{y}) - q(\mathbf{x} - \mathbf{y})| d\mathbf{x} \right) r(\mathbf{y}) dy
$$

For fixed **y**, the inner integral is:

$$\int_{-\infty}^{\infty} |p(\mathbf{x} - \mathbf{y}) - q(\mathbf{x} - \mathbf{y})| d\mathbf{x} = \int_{-\infty}^{\infty} |p(\mathbf{x}) - q(\mathbf{x})| d\mathbf{x}$$

Thus, we obtain:

$$\mathsf{TV}(P * R, Q * R) \leq \frac{1}{2} \int_{-\infty}^{\infty} \left( \int_{-\infty}^{\infty} |p(\mathbf{x}) - q(\mathbf{x})| d\mathbf{x} \right) r(\mathbf{y}) d\mathbf{y}$$

$$\mathsf{TV}(P * R, Q * R) \leq \mathsf{TV}(P, Q)$$

The inequality $\mathsf{TV}(P * R, Q * R) = \mathsf{TV}(P, Q)$ holds if and only if $R$ is a degenerate distribution.

$\square$

## B.1 Proof of Theorem 1

**Theorem 1.** *Let $t + \Delta t$ be the initial noise level. Let $\mathbf{x}_t = \mathsf{ODE}_\theta(\mathbf{x}_{t+\Delta t}, t + \Delta t \to t)$ and $p_t^{\mathsf{ODE}_\theta}$ denote the distribution induced by simulating the ODE with learned drift $s_\theta$. Assume:*
*1. The learned drift $ts_\theta(\mathbf{x}, t)$ is $L_2$-Lipschitz in $\mathbf{x}$, bounded by $L_1$, and $L_0$-Lipschitz in $t$.*
*2. The approximation error $\|ts_\theta(\mathbf{x}, t) - t\nabla \log p_t(\mathbf{x})\| \leq \epsilon_t$.*
*3. All trajectories are bounded by $B/2$.*
*Then, the Wasserstein-1 distance between the generated distribution $p_t^{\mathsf{ODE}_\theta}$ and the true distribution $p_t$ is bounded by:*

$$W_1\left(p_t^{\mathsf{ODE}_\theta}, p_t\right) \leq B \cdot \mathsf{TV}\left(p_{t+\Delta t}^{\mathsf{ODE}_\theta}, p_{t+\Delta t}\right) + e^{L_2 \Delta t} \cdot (\Delta t (L_2 L_1 + L_0) + \epsilon_t) \Delta t$$

*where $\Delta t$ is the step size*

*Proof.* Let $\hat{\mathbf{x}}_t = \mathsf{ODE}_\theta(\mathbf{x}_{t+\Delta t}, t + \Delta t \to t)$ with the corresponding distribution $\hat{p}_t$ and $\mathbf{x}_t = \mathsf{ODE}(\mathbf{x}_{t+\Delta t}, t + \Delta t \to t)$ (simulated under the true score). The proof bounds $W_1\left(p_t^{\mathsf{ODE}_\theta}, p_t\right)$ via triangular inequality:

$$W_1\left(p_t^{\mathsf{ODE}_\theta}, p_t\right) \leq W_1\left(p_t^{\mathsf{ODE}_\theta}, \hat{p}_t\right) + W_1\left(\hat{p}_t, p_t\right) \tag{11}$$

Then we can bound two terms seperately.

1. gradient error: By bounded-diameter inequality,

$$W_1\left(p_t^{\mathsf{ODE}_\theta}, \hat{p}_t\right) \leq B \cdot \mathsf{TV}\left(p_{t+\Delta t}^{\mathsf{ODE}_\theta}, p_{t+\Delta t}\right)$$

2. discretization error: Using Lemma 1 (discretization bound), given $\mathbf{x}_t \sim p_t, \hat{\mathbf{x}}_t \sim \hat{p}_t$

$$\|\hat{\mathbf{x}}_t - \mathbf{x}_t\| \leq e^{L_2 \Delta t} \cdot (\Delta t (L_2 L_1 + L_0) + \epsilon_t) \Delta t$$

where the exponential factor arises from Gronwall's inequality applied to the Lipschitz drift. According to Lemma 5, we can combine terms via triangular inequality:

$$W_1\left(p_t^{\mathsf{ODE}_\theta}, p_t\right) \leq \underbrace{B \cdot \mathsf{TV}\left(p_{t+\Delta t}^{\mathsf{ODE}_\theta}, p_{t+\Delta t}\right)}_{\text{gradient error}} + \underbrace{e^{L_2 \Delta t} \cdot (\Delta t (L_2 L_1 + L_0) + \epsilon_t) \Delta t}_{\text{discretization error}}$$

$\square$

## B.2 Proof of Theorem 2

**Theorem 2** (AdaSDE Error Decomposition). *Consider the same setting as Theorem 1. Let $p_t^{\mathsf{AdaSDE}_\theta}$ denote the distribution after AdaSDE iteration. Then*

$$W_1\left(p_t^{\mathsf{AdaSDE}_\theta}, p_t\right) \leq \underbrace{B \cdot (1 - \lambda(\gamma))\mathsf{TV}\left(p_{t+(1+\gamma)\Delta t}^{\mathsf{AdaSDE}}, p_{t+(1+\gamma)\Delta t}\right)}_{\text{gradient error}}$$

$$+ \underbrace{e^{(1+\gamma)L_2\Delta t}(1+\gamma)\left((1+\gamma)\Delta t\left(L_2 L_1 + L_0\right) + \epsilon_t\right)\Delta t}_{\text{discretization error}}$$

*where $\lambda(\gamma) = 2Q\left(\dfrac{B}{2\sqrt{(t+(1+\gamma)\Delta t)^2 - t^2}}\right).$*

*Proof.* Let $\mathbf{x}_{t+(1+\gamma)\Delta t} \sim p_{t+(1+\gamma)\Delta t}$ and $\hat{\mathbf{x}}_{t+(1+\gamma)\Delta t} \sim p_{t+(1+\gamma)\Delta t}^{\mathsf{AdaSDE}}$. denote exact and generated distributions respectively. And $\bar{\mathbf{x}}_{t+(1+\gamma)\Delta t} \sim p_{t+(1+\gamma)\Delta t}^\theta$. The proof contains three key components:

By Lemma 3, the AdaSDE process contracts the TV distance:

$$\|\bar{\mathbf{x}}_t - \hat{\mathbf{x}}_t\| \leq (1 - \lambda(\gamma))\|\bar{\mathbf{x}}_{t+(1+\gamma)\Delta t} - \hat{\mathbf{x}}_{t+(1+\gamma)\Delta t}\|$$
$$= (1 - \lambda(\gamma))\|\bar{\mathbf{x}}_{t+(1+\gamma)\Delta t} - \mathbf{x}_{t+(1+\gamma)\Delta t}\|$$

Since $\bar{\mathbf{x}}_t \sim p_t^\theta$ and $\hat{\mathbf{x}}_t \sim p_t^{\mathsf{AdaSDE}_\theta}$, we obtain:

$$\mathsf{TV}\left(\bar{p}_t, p_t^{\mathsf{AdaSDE}_\theta}\right) \leq (1 - \lambda(\gamma))\mathsf{TV}\left(\bar{p}_{t+(1+\gamma)\Delta t}, \hat{p}_{t+(1+\gamma)\Delta t}\right)$$
$$= (1 - \lambda(\gamma))\mathsf{TV}\left(\bar{p}_{t+(1+\gamma)\Delta t}, p_{t+(1+\gamma)\Delta t}\right)$$

Using the bounded trajectory assumption $\|\mathbf{x}\| \leq B/2$, we convert TV to Wasserstein-1:

$$W_1\left(\bar{p}_t, p_t^{\mathsf{AdaSDE}_\theta}\right) \leq B \cdot \mathsf{TV}\left(\bar{p}_t, p_t^{\mathsf{AdaSDE}_\theta}\right) \leq B(1 - \lambda(\gamma))\mathsf{TV}\left(\bar{p}_{t+(1+\gamma)\Delta t}, p_{t+(1+\gamma)\Delta t}\right)$$

From Lemma 3, the local ODE error satisfies:

$$\|\mathbf{x}_t^\gamma - \bar{\mathbf{x}}_t^\gamma\| \leq e^{(1+\gamma)L_2\Delta t}(1+\gamma)\left[(1+\gamma)\Delta t(L_2 L_1 + L_0) + \epsilon_t\right]\Delta t$$

According to Lemma 5 and Apply triangle inequality to Wasserstein distances:

$$W_1\left(p_t^{\mathsf{AdaSDE}_\theta}, p_t\right) \leq W_1\left(\bar{p}_t, p_t^{\mathsf{AdaSDE}_\theta}\right) + W_1\left(\bar{p}_t, p_t\right)$$
$$\leq B(1 - \lambda(\gamma))\mathsf{TV}\left(p_{t+(1+\gamma)\Delta t}^{\mathsf{AdaSDE}}, p_{t+(1+\gamma)\Delta t}\right)$$
$$+ e^{(1+\gamma)L_2\Delta t}(1+\gamma)\left[(1+\gamma)\Delta t(L_2 L_1 + L_0) + \epsilon_t\right]\Delta t$$

This completes the error decomposition. $\square$

## B.3 Proof of Theorem 3

**Theorem 3** (TV comparison: AdaSDE vs. ODE). *Assume the same conditions as in Theorem 1 and Theorem 2, and in particular that there exists a compact $K \subset \mathbb{R}^d$ with $diam(K) \leq B$ such that the relevant one-step distributions are supported in $K$. Define*

(i) ODE gradient: $\quad \mathcal{E}_{\text{grad}}^{\mathsf{ODE}} := B \cdot \mathsf{TV}\left(p_{t+\Delta t}^{\mathsf{ODE}_\theta}, p_{t+\Delta t}\right),$

(ii) AdaSDE gradient: $\quad \mathcal{E}_{\text{grad}}^{\mathsf{AdaSDE}} := B\left(1 - \lambda(\gamma)\right)\mathsf{TV}\left(p_{t+(1+\gamma)\Delta t}^{\mathsf{AdaSDE}}, p_{t+(1+\gamma)\Delta t}\right).$

*where $\lambda(\gamma) = 2Q\left(\dfrac{B}{2\sqrt{(t+(1+\gamma)\Delta t)^2 - t^2}}\right) \in (0,1)$ and $B > 0$ is the diameter bound. Then*

$$\mathcal{E}_{\text{grad}}^{\mathsf{AdaSDE}} \leq \mathcal{E}_{\text{grad}}^{\mathsf{ODE}}.$$

*Proof.* By Theorem 1,

$$\mathcal{E}_{\text{grad}}^{\text{ODE}} = B \cdot \text{TV}\Big(p_{t+\Delta t}^{\text{ODE}_\theta}, \, p_{t+\Delta t}\Big).$$

By Theorem 2,

$$\mathcal{E}_{\text{grad}}^{\text{AdaSDE}} = B\left(1 - \lambda(\gamma)\right) \text{TV}\Big(p_{t+(1+\gamma)\Delta t}^{\text{AdaSDE}}, \, p_{t+(1+\gamma)\Delta t}\Big).$$

From $t + \Delta t$ to $t + (1+\gamma)\Delta t$, AdaSDE injects Gaussian noise (a common Markov kernel) into both branches. By Lemma 6 (convolution/pushforward is nonexpansive in TV),

$$\text{TV}\Big(p_{t+(1+\gamma)\Delta t}^{\text{AdaSDE}}, \, p_{t+(1+\gamma)\Delta t}\Big) \;\leq\; \text{TV}\Big(p_{t+\Delta t}^{\text{ODE}_\theta}, \, p_{t+\Delta t}\Big).$$

Since $0 < (1 - \lambda(\gamma)) < 1$, we get

$$\mathcal{E}_{\text{grad}}^{\text{AdaSDE}} = B\left(1 - \lambda(\gamma)\right) \text{TV}\Big(p_{t+(1+\gamma)\Delta t}^{\text{AdaSDE}}, \, p_{t+(1+\gamma)\Delta t}\Big) \;\leq\; B \cdot \text{TV}\Big(p_{t+\Delta t}^{\text{ODE}_\theta}, \, p_{t+\Delta t}\Big) = \mathcal{E}_{\text{grad}}^{\text{ODE}}.$$

$\square$

**Remark 2** (When the inequality is strict)**.** *If $\gamma > 0$, the Gaussian kernel is nondegenerate, and* $\text{TV}(p_{t+\Delta t}^{\text{ODE}_\theta}, \, p_{t+\Delta t}) > 0$ *(equivalently, the two pre-smoothing distributions are not a.e. equal and admit $L^1$ densities), then*

$$\text{TV}\Big(p_{t+(1+\gamma)\Delta t}^{\text{AdaSDE}}, \, p_{t+(1+\gamma)\Delta t}\Big) \;<\; \text{TV}\Big(p_{t+\Delta t}^{\text{ODE}_\theta}, \, p_{t+\Delta t}\Big),$$

*and hence $\mathcal{E}_{\text{grad}}^{\text{AdaSDE}} < \mathcal{E}_{\text{grad}}^{\text{ODE}}$.*

## C  More on AdaSDE

### C.1  Experiment details.

#### Experiment detail in main result

Since AdaSDE has fewer than 40 parameters, its training incurs minimal computational cost. We train $\Theta$ for 10K images, which only takes 5-10 minutes on CIFAR10 with a single 4090 GPU and about 20 minutes on LSUN Bedroom with four 4090 GPUs. For generating reference teacher trajectories, we use DPM-Solver-2 with M=3. For tuning across all datasets, we employed a learning rate of 0.2 along with a cosine learning rate schedule (coslr). The random seed was fixed to 0 to ensure consistent reproducibility of the experimental results. To ensure the robustness of our experimental results, we conducted ten independent runs for each NFE (Number of Function Evaluations) setting on the CIFAR10 dataset. Across these runs, the FID (Fréchet Inception Distance) scores consistently varied by no more than 0.1.

### C.2  Time uniform scheme

[2] proposes a discretization scheme for diffusion sampling given the starting $\sigma_{\max}$, end time $\sigma_{\min}$ and $\epsilon_s$. Denote the number of steps as $N$, then the *time uniform* discretization scheme is:

$$\sigma(t) = \left(e^{0.5\,\beta_d\,t^2 + \beta_{\min}\,t} - 1\right)^{0.5}$$

$$\sigma^{-1}(\sigma) = \frac{\sqrt{\beta_{\min}^2 + 2\,\beta_d\,\ln(\sigma^2 + 1)} - \beta_{\min}}{\beta_d}$$

$$\beta_d = \frac{2\left(\ln\!\left(\sigma_{\min}^2 + 1\right)/\epsilon_s - \ln\!\left(\sigma_{\max}^2 + 1\right)\right)}{\epsilon_s - 1}$$

$$\beta_{\min} = \ln\!\left(\sigma_{\max}^2 + 1\right) - 0.5\,\beta_d$$

$$t_{\text{temp}} = \left(1 + \frac{i}{N-1}\left(\epsilon_s^{1/\rho} - 1\right)\right)^\rho$$

$$t_i = \sigma(t_{\text{temp}})$$

We set $\sigma_{\max} = 80.0$, $\sigma_{\min} = 0.002$, $\rho = 1$ and $\epsilon_s = 10^{-3}$ across all datasets in our experiments.

## C.3 Supplementary experimental results

Table 6: Evaluation on MSCOCO 512×512 (Flux.1-dev).

| Model | NFE | Sampler/Method | FID ↓ | CLIP (%) ↑ |
|---|---|---|---|---|
| **Flux.1-dev 512×512** | 6 | DPM-Solver-2 | 54.09 | 28.49 |
| | | AdaSDE | 35.32 | 29.94 |
| | 8 | DPM-Solver-2 | 30.17 | 29.75 |
| | | AdaSDE | 26.51 | 30.51 |
| | 10 | DPM-Solver-2 | 26.32 | 30.32 |
| | | AdaSDE | 23.54 | 30.77 |

DPM-Solver++(2M)

AdaSDE(Ours)

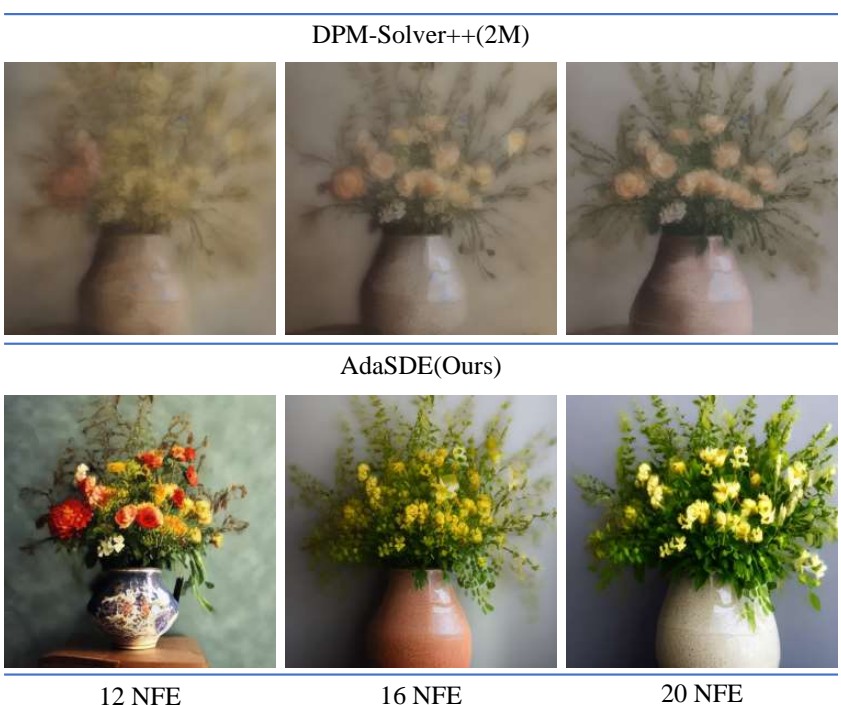

| 12 NFE | 16 NFE | 20 NFE |

Figure 4: Comparison of image synthesis quality under identical NFE constraints using AdaSDE (ours) and DPM-Solver++ (2M). Both methods generate images with Stable Diffusion v1.5 [5] and classifier-free guidance (scale = 7.5) for the prompt *"A photo of some flowers in a ceramic vase"*.

Table 7: Unconditional generation results on CIFAR10 32×32.

| Method | AFS | NFE | | | | | | | |
|---|---|---|---|---|---|---|---|---|---|
| | | 3 | 4 | 5 | 6 | 7 | 8 | 9 | 10 |
| DPM-Solver-v3 | × | - | - | 15.10 | 11.39 | - | 8.96 | - | 8.27 |
| UniPC | × | 109.6 | 45.20 | 23.98 | 11.14 | 5.83 | 3.99 | 3.21 | 2.89 |
| | ✓ | 54.36 | 20.55 | 9.01 | 5.75 | 4.11 | 3.26 | 2.93 | 2.65 |
| DPM-Solver++(3M) | × | 110.0 | 46.52 | 24.97 | 11.99 | 6.74 | 4.54 | 3.42 | 3.00 |
| | ✓ | 55.74 | 22.40 | 9.94 | 5.97 | 4.29 | 3.37 | 2.99 | 2.71 |
| iPNDM | × | 47.98 | 24.82 | 13.59 | 7.05 | 5.08 | 3.69 | 3.17 | 2.77 |
| | ✓ | 24.54 | 13.92 | 7.76 | 5.07 | 4.04 | 3.22 | 2.83 | **2.56** |
| DDIM | × | 93.36 | 66.76 | 49.66 | 35.62 | 27.93 | 22.32 | 18.43 | 15.69 |
| | ✓ | 67.26 | 49.96 | 35.78 | 28.00 | 22.37 | 18.48 | 15.69 | 13.47 |
| DPM-Solver-2 | × | - | 205.41 | - | 45.32 | - | 12.93 | - | 10.65 |
| | ✓ | 227.32 | - | 47.22 | - | 13.68 | - | 10.89 | |
| AMED-Solver | × | - | 17.18 | - | 7.04 | - | 5.56 | - | 4.14 |
| | ✓ | 18.49 | - | 7.59 | - | 4.36 | - | 3.67 | - |
| AdaSDE (ours) | × | - | **10.16** | - | **4.67** | - | **3.18** | - | 2.65 |
| | ✓ | **12.62** | - | **4.18** | - | **2.88** | - | **2.56** | - |

Table 8: Unconditional generation results on ImageNet 64×64.

| Method | AFS | NFE | | | | | | | |
|---|---|---|---|---|---|---|---|---|---|
| | | 3 | 4 | 5 | 6 | 7 | 8 | 9 | 10 |
| UniPC | × | 91.38 | 55.63 | 54.36 | 14.30 | 9.57 | 7.52 | 6.34 | 5.53 |
| | ✓ | 64.54 | 29.59 | 16.17 | 11.03 | 8.51 | 6.98 | 6.04 | 5.26 |
| DPM-Solver++(3M) | × | 91.52 | 56.34 | 25.49 | 15.06 | 10.14 | 7.84 | 6.48 | 5.67 |
| | ✓ | 65.20 | 30.56 | 16.87 | 11.38 | 8.68 | 7.12 | 6.25 | 5.58 |
| iPNDM | × | 58.53 | 33.79 | 18.99 | 12.92 | 9.17 | 7.20 | 5.91 | 5.11 |
| | ✓ | 34.81 | 21.31 | 15.53 | 10.27 | 8.64 | 6.60 | 5.64 | 4.97 |
| DDIM | × | 82.96 | 58.43 | 43.81 | 34.03 | 27.46 | 22.59 | 19.27 | 16.72 |
| | ✓ | 62.42 | 46.06 | 35.48 | 28.50 | 23.31 | 19.82 | 17.14 | 15.02 |
| DPM-Solver-2 | × | - | 140.20 | - | 59.47 | - | 22.02 | - | 11.31 |
| | ✓ | 163.21 | - | 62.32 | - | 23.68 | - | 11.89 | |
| AMED-Solver | × | - | 32.69 | - | 10.63 | - | 7.71 | - | 6.06 |
| | ✓ | 38.10 | - | 10.74 | - | 6.66 | - | 5.44 | - |
| AdaSDE (ours) | × | - | **18.53** | - | **7.01** | - | **5.36** | - | **4.63** |
| | ✓ | **18.51** | - | **6.90** | - | **5.26** | - | **4.59** | - |

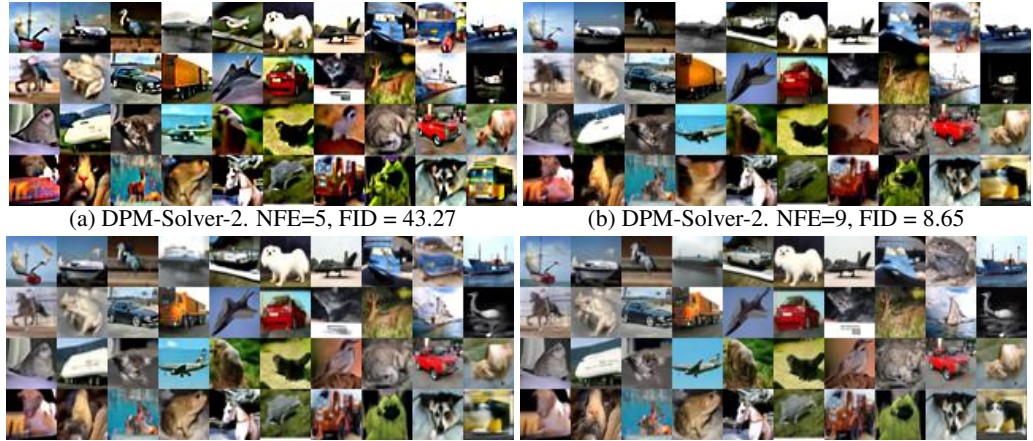

(a) DPM-Solver-2. NFE=5, FID = 43.27          (b) DPM-Solver-2. NFE=9, FID = 8.65

(c) `AdaSDE`. NFE=5, FID = 4.18          (d) `AdaSDE`. NFE=9, FID = 2.56

Figure 5: Qualitative result on CIFAR10 32×32 (5 and 9 NFEs)

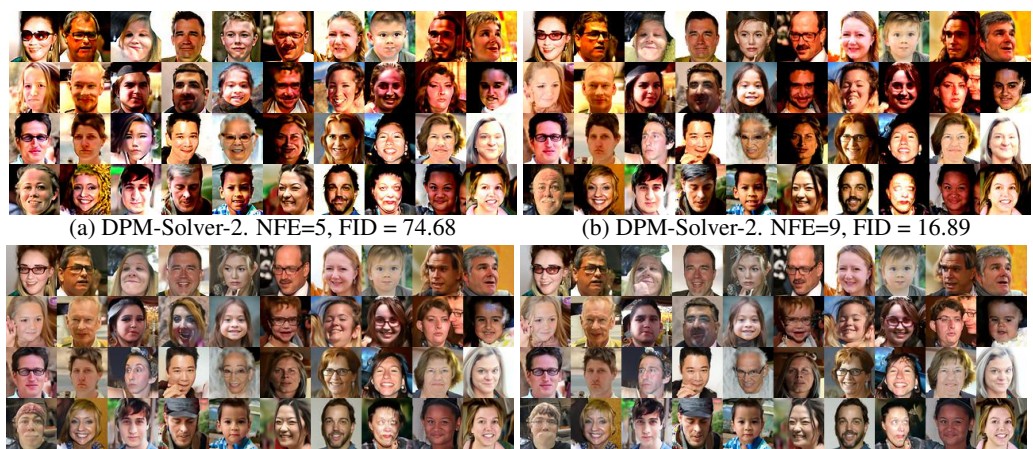

(a) DPM-Solver-2. NFE=5, FID = 74.68          (b) DPM-Solver-2. NFE=9, FID = 16.89

(c) `AdaSDE`. NFE=5, FID = 8.05          (d) `AdaSDE`. NFE=9, FID = 4.19

Figure 6: Qualitative result on FFHQ 64×64 (5 and 9 NFEs)

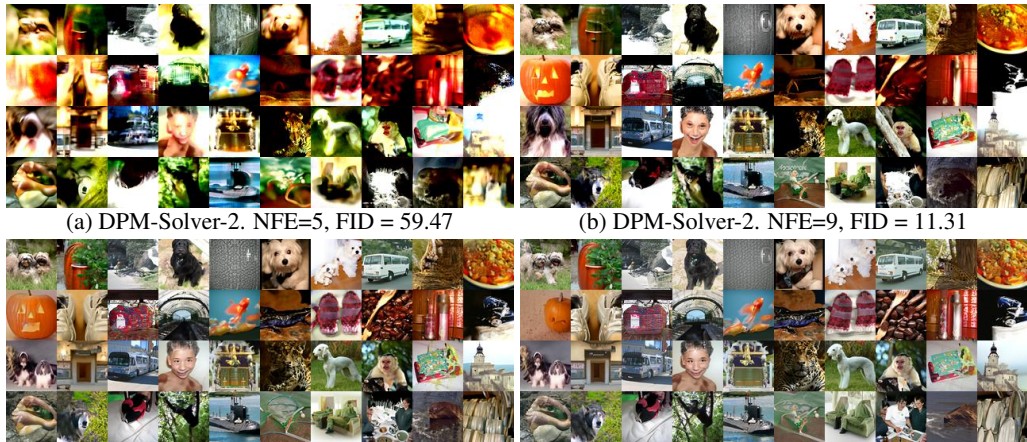

(a) DPM-Solver-2. NFE=5, FID = 59.47          (b) DPM-Solver-2. NFE=9, FID = 11.31

(c) `AdaSDE`. NFE=5, FID = 6.90          (d) `AdaSDE`. NFE=9, FID = 4.59

Figure 7: Qualitative result on ImageNet 64×64 (5 and 9 NFEs)

