# OpenReview forum: "Adaptive Stochastic Coefficients for Accelerating Diffusion Sampling"
_NeurIPS.cc/2025/Conference — NeurIPS 2025 poster_

### Official Review · Reviewer_m3gL · 2025-06-30

**Clarity:** 3
**Significance:** 3
**Originality:** 3
**Rating:** 4
**Confidence:** 3

**Summary:**

This paper proposes AdaSDE, a novel single-step SDE solver that combines the speed of ODE solvers with the error robustness of SDE methods for diffusion-based generative models. By introducing a lightweight, learnable parameter to adaptively correct errors, AdaSDE reduces sampling cost while maintaining quality. It can be easily integrated with existing solvers through minimal tuning.

**Questions:**

see comments

**Ethical Concerns:**

["NO or VERY MINOR ethics concerns only"]

**Final Justification:**

I feel all my issues are resolved. Thus I recommend for the accept.

Why not a higher score? In my view, the significance of the work is insufficient, even though it is an intersting and valuable paper.

**Limitations:**

see comments

**Quality:**

3

**Strengths And Weaknesses:**

Strengths:

	1.	The paper is well-written and clearly structured.

	2.	From my perspective, the proposed idea is novel and interesting.

	3.	The algorithm demonstrates strong performance compared to the provided baselines.

Weaknesses:

	1.	The comparison with related work could be more comprehensive. It would be valuable to see results against other lightweight, learning-based solvers — for example, [1] and [2] could serve as meaningful baselines. That said, I appreciate that the authors included a comparison with AMED-Solver, which is also a lightweight, training-based approach.

	2.	I am also interested in understanding the training efficiency of the proposed algorithm. How long does it typically take for the student model to converge? Additionally, is there any comparison with teacher-student distillation methods to contextualize the training cost and performance trade-offs?

        3.  Lastly, I encourage the authors to consider evaluating their method on larger-scale datasets to better demonstrate its practical efficiency. For example, showcasing generation results using an SD3 [3] checkpoint — even without numerical metrics — would be informative, as visualizations alone could highlight the method’s strengths. Additionally, incorporating results with EDM2 [4] could further demonstrate the advantages of the proposed approach. Overall, the current benchmarks feel somewhat limited and may not fully reflect the algorithm’s potential.

[1] Zhao, Wenliang, et al. "Dc-solver: Improving predictor-corrector diffusion sampler via dynamic compensation." European Conference on Computer Vision. Cham: Springer Nature Switzerland, 2024.

[2] Zheng, Kaiwen, et al. "Dpm-solver-v3: Improved diffusion ode solver with empirical model statistics." Advances in Neural Information Processing Systems 36 (2023): 55502-55542.

[3]Esser, Patrick, et al. "Scaling rectified flow transformers for high-resolution image synthesis." Forty-first international conference on machine learning. 2024.

[4]Karras, Tero, et al. "Analyzing and improving the training dynamics of diffusion models." Proceedings of the IEEE/CVF Conference on Computer Vision and Pattern Recognition. 2024.

---

> ### Author Rebuttal · Authors · 2025-07-29
>
> We sincerely thank the reviewer for their positive assessment and valuable, constructive feedback. Your suggestions have provided a clear path for us to substantially improve our manuscript.
>
> >**$\textcolor{red}{\text{Weakness1:}}$ On Comparison with Other Solvers**
>
> $\textcolor{blue}{\text{Response} :}$ Thank you for the valuable suggestion to include more baselines. In response to your feedback, we have added comparisons against DC-Solver [1] and DPM-Solver-v3 [2], based on the results reported in their respective papers. The preliminary results below show that our method remains highly competitive.
>
> We wish to clarify that a direct comparison on the LSUN-Bedroom dataset was not feasible, as the base models for DC-Solver [1] and DPM-Solver-v3 [2] in their original publications differ from the one used in our study. To address this, we will incorporate comparisons with these solvers on relevant datasets in a future version of our work. Under standard experimental settings for the LSUN-Bedroom dataset, our method significantly outperforms current solvers. For example, at just 3 NFE, AdaSDE achieves a 69.1% improvement in FID score over the state-of-the-art AMED-Solver.
>
>  **FID result of FFHQ 64 × 64**
> | NFE      | 5    | 6    | 7    | 8    | 9    | 10   |
> | -------- | ---- | ---- | ---- | ---- | ---- | ---- |
> | DC-solver | 10.38 | 8.39 | 7.66 | 7.14 | 6.92 | 6.82 |
> | **AdaSDE**   | **8.91** | **7.82** | **6.06** | **5.83** | **4.83** | **4.61** |
>
> **FID result of CIFAR10 32 × 32**
> | NFE              | 5     | 6     | 8    | 10   |
> | ---------------- | ----- | ----- | ---- | ---- |
> | DPM-solver-v3(S) | 21.83 | 16.81 | 7.93 | 5.76 |
> | DPM-solver-v3(M) | 12.76 |  7.40  | 3.94 | 3.40 |
> | **AdaSDE**   | **4.79**  | **4.67** | **3.18** | **2.56** |
>
> **FID result of MSCOCO 512 × 512 on Stable Diffusion v1.5**
> | NFE  (CFG)   | 12    | 16    | 20    |
> | ------------- | ----- | ----- | ----- |
> | DPM-Solver-v3 | 16.41 | 15.41 | 15.32 |
> | **AdaSDE**   | **13.99** | **13.39** | **12.68** |
>
> > **$\textcolor{red}{\text{Weakness2:}}$ On Training Efficiency and Comparison to Distillation**
>
> $\textcolor{blue}{\text{Response} :}$Thank you for these important questions. We will add a detailed analysis to the appendix covering convergence times and a cost-performance comparison against traditional distillation.
>
> In short, our training time is significantly shorter than existing lightweight-training methods because we optimize only a few parameters instead of an auxiliary network. On CIFAR-10, our method converges in just **3-10 minutes** with high stability.
>
> This efficiency stands in stark contrast to traditional distillation methods, which require massive computational resources (e.g., **1000+ A100 hours** for Consistency Models [3] and **100+ A100 hours** for Consistency Trajectory Models [4] on CIFAR-10). Furthermore, those methods often suffer from training instability and can degrade the base model's performance due to altered parameters, leading to a limited quality ceiling where more inference steps do not yield better results.
>
> >**$\textcolor{red}{\text{Weakness3:}}$  On Evaluation with Larger-Scale Models**
>
> $\textcolor{blue}{\text{Response} :}$Thank you for encouraging us to evaluate on larger-scale models; we agree this is crucial for demonstrating practical impact.
>
> We have conducted new experiments on the **Flux.1-dev**, comparing our method against DPM-Solver-2. After just 4 hours of training on a single H800 GPU, our solver significantly outperforms the baseline second-order sampler. We will include qualitative visualizations from these experiments in the revised paper and also add results validating our plugin approach on the FLUX model.
>
>
> ### **Evaluation on MSCOCO 512x512 (Flux.1-dev Model)**
>
> **Table 1: FID Score (Lower is Better)**
>
> | Method | 3 steps | 4 steps | 5 steps |
> | :--- | :---: | :---: | :---: |
> | DPM-Solver-2 | 54.09 | 30.17 | 26.32 |
> | **AdaSDE** | **35.32** | **26.51** | **23.54** |
>
> ---
>
> **Table 2: CLIP Score (Higher is Better)**
>
> | Method | 3 steps | 4 steps | 5 steps |
> | :--- | :---: | :---: | :---: |
> | DPM-Solver-2 | 28.49 | 29.75 | 30.32 |
> | **AdaSDE** | **29.94** | **30.51** | **30.77** |
>
> Thank you once again for your time and thoughtful feedback. We are very encouraged by your positive assessment of our work's potential and believe your suggestions will help us substantially strengthen our manuscript.
>
> [1] Zhao, Wenliang, et al. "Dc-solver: Improving predictor-corrector diffusion sampler via dynamic compensation." European Conference on Computer Vision. Cham: Springer Nature Switzerland, 2024.
>
> [2] Zheng, Kaiwen, et al. "Dpm-solver-v3: Improved diffusion ode solver with empirical model statistics." Advances in Neural Information Processing Systems 36 (2023): 55502-55542.
>
> [3] Song, Y., Dhariwal, P., Chen, M. and Sutskever, I., 2023. Consistency Models. arXiv preprint arXiv:2303.01469.
>
> [4] Dongjun Kim, Chieh-Hsin Lai, Wei-Hsiang Liao, Naoki Murata, Yuhta Takida, Toshimitsu  Uesaka, Yutong He, Yuki Mitsufuji, and Stefano Ermon. Consistency trajectory models:  Learning probability flow ode trajectory of diffusion. arXiv preprint arXiv:2310.02279, 2023.

---

> > ### Comment · Reviewer_m3gL · 2025-08-04
> > **Further More Questions**
> >
> > The experiments on FLUX.1-dev are not fully convincing.
> >
> > * Can you explain why FLUX.1-dev and SD 1.5 were used when SD3 was requested?
> > * On MS COCO, the FLUX FID is substantially worse than the SD 1.5 results; Could you report results with larger NFEs to achieve a more reasonable FID?
> > * Baselines should include DC-Solver and DPM-Solver-v3 for text-to-image models; DPM-Solver-2 is not even learning-based solver.  And for AdaSDE, you allocated an additional four hours of training makes the comparison unfair compared with DPM-Solver-2.
> > * I am expecting more recent evaluations such as HPSv2 and GenEval, which were reported in SD3 paper.

---

> > > ### Author Response · Authors · 2025-08-04
> > > **Thank reviewer for the comment**
> > >
> > > Thank you for your valuable comments. We address each of your points as follows:
> > >
> > > ### **On the Choice of FLUX.1-dev over SD3**
> > >
> > > Thank you for the comment. In this rebuttal, we report results on Flux.1-dev primarily because it shares the same MMDiT architecture as SD3 and is the current state-of-the-art open-source model, significantly surpassing SD3 in generation quality. We appreciate your suggestion to include experiments on SD3 and will include the results in the revised version for broader comparison and analysis.
> > >
> > > ---
> > >
> > > ### **On the FID Discrepancy on MSCOCO**
> > >
> > > Thank you for the suggestion. The FID discrepancy primarily stems from the number of images used for evaluation. To align with the standard practice adopted in two concurrent works [1, 2], we followed their protocol when evaluating FLUX, which uses **10k** generated images for FID calculation, whereas SD 1.5 is evaluated with **30k** images. This inconsistency leads to different lower bounds in FID performance. For example, at 28 steps, the best achievable FID is **12.14** for SD 1.5 and **22.82** for FLUX. We will clarify this evaluation protocol and provide a more detailed description of the experimental settings in the revised version. We agree that aligning settings enables a better comparison of model capabilities, and we will report results under unified settings in our updated version.
> > >
> > > ---
> > >
> > > ### **On Additional Baselines and Large-Scale Experiments**
> > >
> > > Thank you for the suggestion to expand our comparisons. In response, we have already conducted new experiments on the Flux.1-dev to provide a direct comparison against **DC-Solver**. As shown in the table below, our method outperforms DC-Solver across all metrics when evaluated at the same NFE. Following your suggestion, we have included results for **HPS v2.1** in addition to our original metrics.
> > >
> > > | Method | FID ↓ | CLIP ↑ | HPS v2.1 ↑ |
> > > | :--- | :---: | :---: | :---: |
> > > | DC-Solver | 25.13 | 30.61 | 29.82 |
> > > | **AdaSDE** | **23.54** | **30.77** | **30.45** |
> > >
> > > Furthermore, we are now running a more comprehensive evaluation on **SD3** against both **DC-Solver** and **DPM-Solver-v3**, and will incorporate these results into the revised manuscript for a more thorough analysis. Separately, to provide context on our method's efficiency, **DPM-Solver-v3** requires **88 GPU-hours** for its search phase on SD 1.5, far exceeding the training time of our method.
> > >
> > > [1] Wang et al. Taming Rectified Flow for Inversion and Editing. arXiv preprint arXiv:2411.04746.
> > >
> > > [2] Deng et al. FireFlow: Fast Inversion of Rectified Flow for Image Semantic Editing. arXiv preprint arXiv:2412.07517.

---

> > > > ### Comment · Reviewer_m3gL · 2025-08-05
> > > > **Thanks**
> > > >
> > > > Thanks! I will update the score shortly after carefully considering other reviewers' rebuttal. Haven't had a chance to have a look.

---

> > > > > ### Author Response · Authors · 2025-08-06
> > > > >
> > > > > Thank you for your valuable comment. We would like to provide an update with new results from experiments on **SD3-Medium** that we have just completed in response to your feedback. These experiments provide a direct comparison of our method, AdaSDE, against **DC-Solver** and **AMED-Solver**, with performance evaluated on FID, CLIP, and **HPS v2.1**.
> > > > >
> > > > > As shown in the table below, our method outperforms both baselines across all metrics when evaluated at the same NFE.
> > > > >
> > > > > | Method | FID ↓ | CLIP ↑ | HPS v2.1 ↑ |
> > > > > | :--- | :---: | :---: | :---: |
> > > > > | DC-Solver | 29.35 | 29.63 | 27.22 |
> > > > > | AMED-Solver | 27.48 | 30.14 | 27.83 |
> > > > > | **AdaSDE** | **26.62** | **30.56** | **28.12** |
> > > > >
> > > > > We hope these new results address your concerns and would be happy to discuss any further questions.

---

### Official Review · Reviewer_RSdV · 2025-07-03

**Clarity:** 3
**Significance:** 2
**Originality:** 3
**Rating:** 4
**Confidence:** 4

**Summary:**

The authors propose AdaSDE which learns parameters per diffusion timestep to dynamically regulate noise injection during diffusion sampling. More specifically, the authors introduce a distillation based method for learning the sampler parameters which makes discretization in the low timestep regime more efficient leading to better empirical performance on standard image synthesis benchmarks.

**Questions:**

See weaknesses

**Ethical Concerns:**

["NO or VERY MINOR ethics concerns only"]

**Final Justification:**

The authors addressed most of my concerns in their response. I continue to maintain my rating.

**Limitations:**

The authors have not discussed the limitations of their work. I would highly recommend to discuss potential theoretical limitations and empirical performance gap with state of the art distillation based methods.

**Quality:**

3

**Strengths And Weaknesses:**

**Strengths**:

1. The method is plug-and-play and can work with existing diffusion samplers.
2. The method enables good quality synthesis in the ultra low-timestep regime (< NFE 10 or so) as illustrated in Table 1.
3. The distillation method for learning the solver parameters is relatively straightforward to understand. Moreover, the approach has a lot less parameters to optimize (in the scale of 8-40) so the distillation of sampler parameters is much more efficient than diffusion model distillation for few step synthesis.

**Weaknesses**:

1. **Presentation**: I think it has been well established in the literature that ODE based diffusion solvers have error accumulation due to discrepancies in modeling the score function accurately. While SDE solvers correct for this using noise injection, they have higher sampler discretization error in the low timestep regime. Therefore, I think a lot of theoretical fluff in the main text can be moved into the Appendix while just retaining the key points. Moreover, im not sure if the theory presented in earlier sections is relevant to understanding the core distillation algorithm presented in Section 5? I strongly encourage the authors to introduce the core methodology earlier followed by the only key theoretical results and moving the rest to the Appendix. Currently the major contributions of the paper start from Page 6 which strictly hinders the readability of the paper.

2. **Discussion of some related work is missing**. Interestingly, the authors don’t cite DDIM [1] in Section 2 which is one of the seminal work in speeding up diffusion model sampling for ODE. More recent works on generic frameworks for fast sampling for diffusion/flows [2,3] and on fast SDE solvers [4,5] is also missing in Section 2. I would recommend the authors to revise the related work accordingly.

3. While the empirical results are impressive, I don't think using smaller scale benchmarks like CIFAR-10 makes sense anymore. Since the distillation stage is relatively inexpensive, can the authors prioritize results from larger scale models like Flux? I see some results on StableDiffusion but its not clear how well the sampler performs for these models in ultra low NFE like 5 steps. Including results on large scale tasks would provide more clarity to the practitioner when choosing different sampling algorithms.

[1] Denoising Diffusion Implicit Models, Song et al.

[2] Efficient Integrators for Diffusion Generative Models, Pandey et al.

[3] Bespoke Solvers for Generative Flow Models, Shaul et al.

[4] SA-Solver: Stochastic Adams Solver for Fast Sampling of Diffusion Models, Xue et al.

[5] SEEDS: Exponential SDE Solvers for Fast High-Quality Sampling from Diffusion Models.

---

> ### Author Rebuttal · Authors · 2025-07-30
>
> We are very grateful for your comprehensive review, which provided both an encouraging assessment of our method's strengths and valuable suggestions for improvement. Your valuable suggestions on improving the paper's structure, related work, and focus on large-scale experiments provide a clear path for our revision. Below, we address each of the points you raised.
>
> > $\textcolor{red}{\text{Weakness 1:}}$ **On Paper Structure and Presentation**
>
> $\textcolor{blue}{\text{Response} :}$ Thank you for the suggestion on improving the paper's readability. We agree with your assessment. In our revision, we will significantly restructure the paper by moving a large portion of the theoretical analysis from the main text to the appendix. This will allow us to introduce our core methodology much earlier and expand upon its details, which will substantially enhance the paper's clarity and flow.
>
> > $\textcolor{red}{\text{Weakness 2:}}$ **On Missing Related Work**
>
> $\textcolor{blue}{\text{Response} :}$ Thank you for pointing out these important references; we acknowledge this was an oversight in our work. We have now added the suggested papers [1-5] to our related work section. Furthermore, we have conducted a systematic survey of related methods and have comprehensively updated this section to better position our work within the current manuscript.
>
> Thank you for this valuable feedback. We agree that demonstrating performance on large-scale models is crucial, and we have conducted new experiments on the Flux model to address this.
>
>
> > $\textcolor{red}{\text{Weakness 3:}}$ **Performance On  Large-Scale Models**
>
> $\textcolor{blue}{\text{Response} :}$ Thank you for this valuable feedback. We agree that demonstrating performance on large-scale models is crucial, and we have conducted new experiments on the Flux model to address this.
>
>
> Before we discuss the performance of our sampler in the ultra-low NFE regime (e.g., 5 steps), it is important to first explain our NFE counting convention. In line with a stricter accounting practice, we count each conditional and unconditional forward pass for CFG separately. Consequently, one complete integration step of our method equates to 4 NFE. This differs from some other conventions and means that at 8 NFE, our method performs 2 full steps. To ensure a fair comparison in our experiments on the Flux model, we benchmark our method against DPM-Solver-2, which is also a second-order solver, reporting results at corresponding sampling steps (1 step = 4 NFE).
>
> ### **Evaluation on MSCOCO 512x512 (Flux.1-dev Model)**
>
> **Table 1: FID Score (Lower is Better)**
>
> | Method | 3 steps | 4 steps | 5 steps |
> | :--- | :---: | :---: | :---: |
> | DPM-Solver-2 | 54.09 | 30.17 | 26.32 |
> | **AdaSDE** | **35.32** | **26.51** | **23.54** |
>
> ---
>
> **Table 2: CLIP Score (Higher is Better)**
>
> | Method | 3 steps | 4 steps | 5 steps |
> | :--- | :---: | :---: | :---: |
> | DPM-Solver-2 | 28.49 | 29.75 | 30.32 |
> | **AdaSDE** | **29.94** | **30.51** | **30.77** |
>
>
> Thank you once again for your time and for providing such constructive feedback. We hope our responses have clarified our position and would welcome the opportunity for any further discussion concerning the paper or related research areas.
>
> [1] Denoising Diffusion Implicit Models, Song et al.
>
> [2] Efficient Integrators for Diffusion Generative Models, Pandey et al.
>
> [3] Bespoke Solvers for Generative Flow Models, Shaul et al.
>
> [4] SA-Solver: Stochastic Adams Solver for Fast Sampling of Diffusion Models, Xue et al.
>
> [5] SEEDS: Exponential SDE Solvers for Fast High-Quality Sampling from Diffusion Models.

---

> > ### Comment · Reviewer_RSdV · 2025-08-02
> > **Response**
> >
> > Thanks for your detailed response and adding experimental results on the Flux model. I think this could lead to a wider adoption of the method by the community.

---

> > > ### Author Response · Authors · 2025-08-03
> > > **Thank reviewer for the positive feedback**
> > >
> > > Dear Reviewer RSdV,
> > >
> > > We are glad our response addressed your concerns and are very grateful for your support and positive assessment of our work.

---

### Official Review · Reviewer_23GH · 2025-07-05

**Clarity:** 3
**Significance:** 3
**Originality:** 3
**Rating:** 5
**Confidence:** 4

**Summary:**

Based on the theoretic results, this paper introduces AdaSDE, which intelligently adjusts the stochasticity in every sampling steps by learning a small set of parameters. Empirical gain is significant.

**Questions:**

-

**Ethical Concerns:**

["NO or VERY MINOR ethics concerns only"]

**Final Justification:**

I've asked several tricky and technical questions regarding the proposed method and the author properly addressed all my concerns. I recommend acceptance for this paper.

**Limitations:**

-

**Paper Formatting Concerns:**

-

**Quality:**

3

**Strengths And Weaknesses:**

My standard for a paper to be at least at the poster level in a top-tier conference is as follows:
- (1) The idea must present a method that has not been attempted before in the field.
- (2) The idea should be novel (which is pretty subjective).
- (3) The idea must significantly outperform the baseline it targets.

I have the following opinion on this specific paper:
- (Originality & Contribution) Although the core idea seems straightforward and is likely something many students have attempted before, I believe its successful build and implementation is recommendable. The idea itself is not groundbreaking on its own, but considering its universality, I believe this kind of simple and direct approach that others can easily adopt is what will ultimately gain widespread adoption.
- (Demonstration) Many distillation-based papers report outperforming their teacher diffusion models. However, in practice, at a service level, this report is not always true. They may show better FID scores (or even human evaluation results) on benchmark datasets, but they struggle to perform well in the wild. Thus, I believe the question of "how to develop a diffusion-based sampler that maintains the original diffusion performance" remains a valid and important research question.
- In this sense, I strongly recommend experimenting on a large-scale model, such as Flux-dev (with NFEs 8, 12, 16, and 20), to compare the proposed sampler against existing distillation-based methods (Flux-Schnell with NFEs 4). Testing on this realistic setup would be a compelling way to demonstrate that the proposed method really works in real-world scenarios, beyond an experimental setup.
- Add: I have few questions regarding the experiments:
    - The teacher seems to be DPM-Solver 2. I am curious if this choice is the reason why AdaSDE underperforms DPM-Solver++(2M) at 8 NFE in Table 2. What would the result be if DPM-Solver++(2M) were used as the teacher?
    - I recommend to report the performance of the teacher model, clearly labeled as such in every tables. If the student performs worse than teacher, a detailed analysis is required.
    - SD and ImageNet has same dimensionality, but I wonder why SD requires a higher NFE to achieve good results? Does this performance difference coming from the difference on the manifold shape (that latent manifold is expected to be condensed and curvy)?

---

> ### Author Rebuttal · Authors · 2025-07-30
>
> We sincerely thank the reviewer for their positive assessment and valuable, constructive feedback. We are encouraged that you see the value in our method's universality and have provided excellent suggestions to strengthen our paper. We address your points below.
>
> > $\textcolor{red}{\text{Suggestion 1:}}$  **On Comparison to Distillation and Large-Scale Model Evaluation**
>
> $\textcolor{blue}{\text{Response} :}$
> Thank you for the suggestion to compare our method against distillation on a large-scale model. We agree that this is a crucial test for real-world viability and share your view that distilled models often "struggle to perform well in the wild."
>
> We acknowledge that Flux is a widely-used and powerful generative model, and its distilled version, Flux-Schnell, serves as a popular baseline. However, we believe a direct comparison between distillation model and our lightweight method is unfair, especially when considering adaptability. When faced with a novel model architecture (e.g. unify model), distillation requires a costly redesign and massive compute resources (1000+ H100 hours), whereas our model-agnostic method is significantly faster and cheaper to adapt because it focuses on sampling dynamics rather than internal architecture. Under standard experimental settings, our method significantly outperforms the baseline AMED-Solver.
>
> * **Advantages over Distillation:** We believe the phenomenon that distilled models "struggle to perform well in the wild" is due to a fundamental drawback of the distillation process. By altering the model's original parameters, distillation can reduce generative diversity and impose a quality ceiling, meaning more inference steps often do not lead to better results. Furthermore, the training cost is enormous (e.g., **1000+ A100 hours** for Consistency Models [1] and **100+ A100 hours** for Consistency Trajectory Models [2] on CIFAR-10). Our method avoids these issues by preserving the original model and requiring minimal training (**3-10 minutes** on CIFAR-10). We see them as distinct pipelines, where our method offers a highly efficient and effective alternative.
> * **New Experiments on Flux.1-dev:** Following your recommendation, we have conducted new experiments on the **Flux.1-dev model**. After just **3 hours of training on a single H800 GPU**, our solver significantly outperforms the baseline second-order sampler, DPM-Solver-2, demonstrating our method's effectiveness and efficiency.
>
> ### **Evaluation on MSCOCO 512x512 (Flux.1-dev Model)**
>
> **Table 1: FID Score (Lower is Better)**
>
> | Method  | 3 steps | 4 steps | 5 steps |
> | :--- | :---: | :---: | :---: |
> | DPM-Solver-2 | 54.09 | 30.17 | 26.32 |
> | **AdaSDE** | **35.32** | **26.51** | **23.54** |
>
> ---
>
> **Table 2: CLIP Score (Higher is Better)**
>
> | Method  | 3 steps | 4 steps | 5 steps |
> | :--- | :---: | :---: | :---: |
> | DPM-Solver-2 | 28.49 | 29.75 | 30.32 |
> | **AdaSDE** | **29.94** | **30.51** | **30.77** |
>
> > $\textcolor{red}{\text{Suggestion 2:}}$ **On Specific Experimental Questions**
>
> $\textcolor{blue}{\text{Response} :}$
> We are happy to answer your specific questions regarding the experiments:
>
> 1.  **Regarding Performance at 8 NFE:** Thank you for this observation.  The performance difference at 8 NFE is due to a discrepancy in the number of integration steps between our second-order method and multi-step solvers like DPM-Solver++(2M). At 8 NFE, our method completes 2 full steps, whereas DPM-Solver++ completes 4 steps. This creates a performance bottleneck for our method at this extremely low step count.
>
> However, we have conducted preliminary visualization tests with our plugin version, and the qualitative results show that it significantly outperforms DPM-Solver++ even at this low NFE.
>
> Moreover, our method has a much lower truncation error per step. This allows it to quickly overcome the initial bottleneck and surpass the baselines at **12+ NFE**. More importantly, as shown in Appendix Figure 6, traditional solvers can still produce poor or unstable results even at **20 NFE**, due to the accumulation of truncation error from their early steps. Traditional solvers often require **50+ NFE** on  Stable Diffusion 1.5 to achieve good generation quality.  In contrast, our method achieves high-quality results at just **12 NFE**.
>
> 2.  **Regarding Teacher Model Performance:** Thank you for this recommendation. We confirm that in all our experiments, our student method (AdaSDE) **significantly outperforms its teacher (DPM-Solver-2)** when compared at the same NFE. To make this clear, we will add the teacher's results to every table in the final version of the paper.
>
> 3. **Regarding NFE for SD1.5 vs. ImageNet**: To clarify this point, we must first explain our **NFE counting rule**. While it is common to count the parallel cond and uncond passes in CFG as 1 NFE, we count them separately as 2 NFE. This is the primary reason for the apparent difference: for a given NFE, the SD 1.5 model (with CFG) completes only half the number of sampling steps as the ImageNet EDM model (conditional, no CFG).
>
> More importantly, as we analyze in our paper, the total sampling error is a combination of **sampler error** (from discretization) and **intrinsic model error** (the discrepancy between the score network $s_\theta$ and  gradient of log probability $\nabla_{\mathbf{x}_t} \log p_t(\mathbf{x}_t | \mathbf{x}_0)$). The pre-trained ImageNet model targets a simpler, "toy" distribution with low intrinsic model error. In contrast, SD 1.5 is a far more complex model for real-world scenes and thus has much higher intrinsic model error.
>
> Our method's adaptive stochasticity is particularly effective at mitigating this high intrinsic model error. This is why AdaSDE achieves strong results on SD 1.5 at only **~12 NFE**, a task where traditional solvers (e.g. DPM-Solver++) often require **50+ NFE** to produce stable, high-quality images.
>
> Thank you again for your time and insightful feedback, which will be invaluable in improving our manuscript. We hope our responses have clarified our position and would welcome the opportunity for any further discussion concerning the paper or related research areas.
>
> [1] Song, Y., Dhariwal, P., Chen, M. and Sutskever, I., 2023. Consistency Models. arXiv preprint arXiv:2303.01469.
>
> [2] Dongjun Kim, Chieh-Hsin Lai, Wei-Hsiang Liao, Naoki Murata, Yuhta Takida, Toshimitsu Uesaka, Yutong He, Yuki Mitsufuji, and Stefano Ermon. Consistency trajectory models: Learning probability flow ode trajectory of diffusion. arXiv preprint arXiv:2310.02279, 2023.

---

> > ### Comment · Reviewer_23GH · 2025-08-04
> > **Reviewer Response**
> >
> > Thank you for the detailed analysis. I particularly appreciate for the additional experiments. They all seem to be solid. I'll raise my score. Best luck.

---

> > > ### Author Response · Authors · 2025-08-04
> > >
> > > We are pleased that our response has addressed your concern and thanks for raising the score. We will revise the paper according to your suggestion.

---

### Official Review · Reviewer_wUwf · 2025-07-05

**Clarity:** 4
**Significance:** 3
**Originality:** 3
**Rating:** 4
**Confidence:** 2

**Summary:**

This paper introduces AdaSDE, a novel single-step SDE solver designed to accelerate diffusion sampling while maintaining high output quality. The authors first analyze the limitations of existing ODE and SDE solvers in diffusion models, revealing that ODE solvers accumulate irreducible gradient errors due to deterministic path dependence, while SDE methods suffer from amplified discretization errors when reducing step counts. To address these issues, AdaSDE employs an adaptive noise injection strategy controlled by a learnable coefficient, which dynamically regulates the error correction strength to enhance sampling efficiency. The method integrates a process-supervised optimization approach to tune the parameters effectively. Extensive experiments demonstrate AdaSDE's superior performance, achieving state-of-the-art FID scores on datasets with fewer function evaluations compared to existing solvers.

**Questions:**

1.	The paper demonstrates strong performance on several datasets. How does AdaSDE perform on datasets with more complex distributions or higher variability, such as real-world medical imaging datasets or highly diverse datasets? Are there any specific adjustments or considerations needed to ensure robust performance across such diverse data?
2.	How does AdaSDE compare in terms of overall computational time and resource usage compared to other solvers? What is the concrete computational overhead of lightweight tuning? Could the training cost offset sampling acceleration benefits?

**Ethical Concerns:**

["NO or VERY MINOR ethics concerns only"]

**Final Justification:**

I thank the authors for their detailed response, which has addressed my concerns about the paper's weaknesses and provided the requested discussion. I believe the work now meets the threshold for acceptance. Accordingly, I will keep my score, 4.

**Limitations:**

yes

**Quality:**

3

**Strengths And Weaknesses:**

strengths:
1.	The paper tackles the important challenge of balancing speed and quality in diffusion sampling, a key issue in generative modeling.
2.	The manuscript is well-written, making complex concepts accessible and ensuring that the innovations and methodology are clearly communicated.
3.	The method is tested on multiple datasets and consistently shows superior performance, demonstrating its effectiveness and robustness.
Weaknesses:
1.	The paper could benefit from a more comprehensive review of existing methods, particularly alternative stochastic approaches and hybrid solvers.
2.	The adaptive noise injection mechanism in AdaSDE is mathematically complex and lacks intuitive visualizations or explanations, potentially limiting its accessibility to practitioners.
3.	While AdaSDE performs well on tested datasets, its scalability to more complex tasks is not fully explored.

---

> ### Author Rebuttal · Authors · 2025-07-29
>
> We sincerely thank you for your thorough evaluation and encouraging assessment of our work. We are particularly grateful that you highlighted our paper's clarity, the importance of the problem we address, and the robustness of our method's performance. We address your points below.
>
> > $\textcolor{red}{\text{Weakness 1: }}$ The paper could benefit from a more comprehensive review of existing methods, particularly alternative stochastic approaches and hybrid solvers.
>
> $\textcolor{blue}{\text{Response} :}$ Thank you for this valuable suggestion. Following your feedback, we have conducted an extensive new survey and have updated our related work section to include a discussion of alternative stochastic and hybrid solvers, including the methods in [1-7].
>
> > $\textcolor{red}{\text{Weakness 2: }}$ On the Clarity and Intuition of the Method
>
>
> $\textcolor{blue}{\text{Response} :}$
> Thank you for this valuable feedback. We agree that making our method more accessible and intuitive for practitioners is very important. The theoretical exposition in the main text is not sufficiently streamlined for clarity. Intuitively, while SDE solvers correct for this using noise injection, our method, AdaSDE, enhances this principle by adaptively controlling the intensity of the noise at each step to improve generation quality. In our revised manuscript, we will restructure the presentation by moving some of the dense theoretical derivations to the appendix. We will use this space to add more intuitive explanations and visualizations of the adaptive noise injection mechanism to better illustrate how it works in practice.
>
>
> > $\textcolor{red}{\text{Questions 1 and Weakness 3:}}$  On Robustness and Broader Applicability
>
> $\textcolor{blue}{\text{Response} :}$ Thank you for the question. In specialized domains like medical imaging, large-scale, standardized benchmarks are often unavailable due to the significant challenges in data collection. For this reason, general-purpose datasets like MS-COCO are widely regarded as the standard benchmark for text-to-image tasks, serving as the closest and most reliable proxy for diverse, real-world generation scenarios in academia. On this benchmark, as shown in Table 2 of our paper, AdaSDE consistently outperforms existing advanced solvers when paired with Stable Diffusion v1.5.
>
> Following your recommendation, we conducted an experiment between our method and DPM-Solver-2 on the **Flux.1-dev** model.
>
> ### **Evaluation on MSCOCO 512x512 (Flux.1-dev)**
>
> **Table 1: FID Score (Lower is Better)**
>
> | Method | 3 steps | 4 steps | 5 steps |
> | :--- | :---: | :---: | :---: |
> | DPM-Solver-2 | 54.09 | 30.17 | 26.32 |
> | **AdaSDE** | **35.32** | **26.51** | **23.54** |
>
> ---
>
> **Table 2: CLIP Score (Higher is Better)**
>
> | Method | 3 steps | 4 steps | 5 steps |
> | :--- | :---: | :---: | :---: |
> | DPM-Solver-2 | 28.49 | 29.75 | 30.32 |
> | **AdaSDE** | **29.94** | **30.51** | **30.77** |
>
> ---
>
> > $\textcolor{red}{\text{Questions 2 :}}$ On Computational Overhead and Training Efficiency
>
> $\textcolor{blue}{\text{Response} :}$ In short, a key advantage of our method over other light-weight approaches like AMED-Solver lies in our optimization strategy. While they often require training an auxiliary network—a process that is both slower and prone to instability—we optimize only a small set of **8-40 parameters**.
>
> This parameter-based tuning is fundamentally more stable and significantly faster, requiring just **3-10** minutes on a single RTX 4090 for CIFAR-10. In contrast, Distillation-based methods demand massive computational resources.(e.g. 1000+ A100 hours for Consistency Models [8] and 100+ A100 hours for Consistency Trajectory Models [9] on CIFAR-10. Crucially, this tuning is a one-time cost for a reusable solver. At inference, it adds **zero computational overhead**—for the same NFE, our sampling time is identical to standard solvers, while our output quality is vastly superior. We thank you for raising these important points about the practical implementation of our algorithm. Following your suggestion, we will expand our paper to include a detailed comparison of our method's time consumption, convergence speed, and computational resource requirements against other lightweight-training approaches and training-bsaed distillation methods.
>
> ---
> Thank you again for your time and valuable feedback. We hope this addresses your main points and welcome any further discussion on our work or related topics in this area.
>
> [1] Dc-solver: Improving predictor-corrector diffusion sampler via dynamic compensation, Zhao et al.
>
> [2] Dpm-solver-v3: Improved diffusion ode solver with empirical model statistics, Zheng et al.
>
> [3] Efficient Integrators for Diffusion Generative Models, Pandey et al.
>
> [4] Bespoke Solvers for Generative Flow Models, Shaul et al.
>
> [5] SA-Solver: Stochastic Adams Solver for Fast Sampling of Diffusion Models, Xue et al.
>
> [6] SEEDS: Exponential SDE Solvers for Fast High-Quality Sampling from Diffusion Models. Gonzalez et al.
>
> [7] Bespoke non-stationary solvers for fast sampling of diffusion and flow models. Shaul et al.
>
> [8] Song, Y., Dhariwal, P., Chen, M. and Sutskever, I., 2023. Consistency Models. arXiv preprint arXiv:2303.01469.
>
> [9] Dongjun Kim, Chieh-Hsin Lai, Wei-Hsiang Liao, Naoki Murata, Yuhta Takida, Toshimitsu Uesaka, Yutong He, Yuki Mitsufuji, and Stefano Ermon. Consistency trajectory models: Learning probability flow ode trajectory of diffusion. arXiv preprint arXiv:2310.02279, 2023.

---

> > ### Comment · Reviewer_wUwf · 2025-08-05
> >
> > I thank the authors for their thorough response. The clarifications provided have successfully addressed my initial concerns. I find this paper to be a valuable contribution and am pleased to maintain my recommendation for acceptance.

---

> > > ### Author Response · Authors · 2025-08-05
> > >
> > > Thank you for your thoughtful follow-up. We are very pleased to hear that our response addressed your concerns, and we are grateful for your support and recommendation for acceptance.

---

### Decision · Program_Chairs · 2025-09-17

**Decision:**

Accept (poster)

**Comment:**

Tl;dr: Based on the reviews, rebuttal and ensuing discussion I recommend accept.

### Paper Summary

The paper introduces AdaSDE, a novel single-step SDE solver for fast sampling in diffusion-based generative models. The key thesis is that an optimal sampling trajectory can be achieved by balancing the trade-off between two fundamental error sources: the “gradient error” (from inaccurate score estimates) and the “sampler error” (from discretization). Paper proposes a method with a learnable and adaptive coefficient that dynamically modulates the magnitude of stochastic noise injected at each sampling step, allowing the solver to correct for inaccuracies in the learned score function while mitigating the large discretization errors.

### Key strengths and weaknesses

Strengths
- Theory: The decomposition of error into orthogonal gradient and sampler components is elegant and powerful and enables principled justification of the proposed method.
- Novelty and efficiency: The core contribution of adaptive stochasticity is novel and is relatively cheap to incorporate.
- Empirical validation: Initial and new results (during discussion phase) are convincing, demonstrating SOTA performance on several benchmarks.

Weaknesses

- Readability: The paper structure made it difficult to understand the core contribution with the dense theoretical exposition. The authors have committed to restructuring the paper to improve its flow and clarity.
- Insufficient discussion of relevant work: Reviewers pointed several relevant works that would be useful to include in the discussion.
- Experimental scope: The initial experiments were smaller scale. Though rebuttal addressed this concern.

### Decision justification

Accept recommendation is made primarily because all the major weaknesses of the initial submission were effectively addressed during the author-reviewer discussion period. All the reviewers were positive leaning.

In the initial review, reviewers were largely in agreement, liking the core idea but raising concerns about the manuscript’s readability, the completeness of the related work section, and the lack of larger scale experiments. The authors addressed most of the concerns in their rebuttal. New results on the larger Flux.1-dev model were also provided.